# Solution-sheared supramolecular oligomers with enhanced thermal resistance in interfacial adhesion and bulk cohesion

Gang Lu [1,2] ✉, Rui Ma[3], Yuanyuan Zhao [4], Dianyu Wang [5] ✉,
Wentao Shang[2,6], Huaguo Chen[7], Shahid Ali Khan[2], Ming Li[8] ✉ & Eduardo Saiz[8]

Developing strong, thermally resistant adhesives for load-bearing applications remains challenging. Here we report a class of solution-sheared supramolecular oligomers that exhibit exceptional adhesive strength and toughness across a broad temperature range. These adhesives achieve a debonding work up to 23.6 kN/m and a lap shear strength exceeding 30.6 MPa, surpassing commercial structural adhesives by up to eightfold on metal and glass surfaces. Impressively, they retain a lap shear strength above 21 MPa even at 120 °C, outperforming current leading commercial alternatives. This performance arises from hierarchical nanostructures formed during solution shearing, which create enlarged, ordered nanocrystals and aligned nanofibrils within the bulk, enhancing mechanical robustness and toughness. Simultaneously, hydrogen-bonded nanocrystals anchored at the surface significantly strengthen interfacial adhesion. This multiscale structural organization enables thermal tolerance, crack resistance, and efficient energy dissipation, setting a new paradigm for high-performance, reusable adhesives capable of multiple rebonding cycles. Our work demonstrates how solution-shearing simultaneously optimizes adhesion chemistry and multiscale nano/microstructural control, achieving synergistic improvements in interfacial adhesion and bulk cohesion.

Structural adhesives, also called load-bearing adhesives, are integral to numerous industrial and biomedical applications, spanning sectors from construction and automotive to aerospace and wound care[1,2]. Commercial structural adhesives, predominantly comprising cyanoacrylates, epoxies, methacrylates, and polyvinyl alcohol systems, can establish strong adhesion to target surfaces. However, their irreversible curing mechanism prevents reuse or recycling[3,4], significantly restricting sustainable applications where reversible adhesion is

required. In response, there is a pressing need to develop a new class of structural adhesives able to exhibit strong and repeatable adhesion.

Hydrogen-bonded supramolecular polymer adhesives have shown significant advantages due to their robust and reversible adhesion, tunable mechanical properties, and stimuli-responsive capacities[5–14]. Robust bonding necessitates strong interfacial joining, typically involving a mix of hydrogen bonding and various interactions with the substrates[15–19]. Additionally, tough adhesive layers are

[1]Department of Chemical and Biomolecular Engineering, University of Pennsylvania, Philadelphia, PA, USA. [2]School of Energy and Environment, City University of Hong Kong, Hong Kong, China. [3]NTNU Nanomechanical Lab, Department of Structural Engineering, Norwegian University of Science and Technology, Trondheim, Norway. [4]School of Fashion and Textile, The Hong Kong Polytechnic University, Hong Kong, China. [5]School of Chemical Engineering, Zhengzhou University, Zhengzhou, China. [6]Energy and Electricity Research Center, International Energy College, Jinan University, Guangdong, China. [7]Department of Architecture and Civil Engineering, City University of Hong Kong, Hong Kong, China. [8]Centre of Advanced Structural Ceramics, Department of Materials, Imperial College London, London, UK. ✉e-mail: ganglu7@seas.upenn.edu; wangdy@zzu.edu.cn; mingli24@hku.hk

essential for promoting energy dissipation within the bulk[10,20–23]. For example, Liu et al. demonstrated a fatigue-resistant PVA hydrogel adhesive that shows remarkable toughness and a high fatigue threshold of 800 J/m$^2$, attributed to ordered nanocrystalline domains anchored at the interfaces[24]. The mechanical toughness of supramolecular adhesives usually arises from mechanisms, such as the cleavage of sacrificial bonds, unraveling of entangled chains, and hysteresis within the polymer networks[21,25–27]. For instance, Sun and colleagues developed a dual crosslinked hydrogel that showcased high mechanical toughness and stretchability, where an alginate-calcium ion network facilitated energy dissipation during bulk deformation, achieving a fracture energy of 9000 J/m$^2$ [28]. While design and optimization of interfacial adhesion and bulk cohesion have led to significant advancements in enhancing the robustness and toughness of hydrogen-bonded supramolecular polymers[10,20,29,30], these efforts are often impeded by a limited fundamental understanding of the interplay between interfacial adhesion, adhesive chemistry, and hierarchical structures spanning from nano- to microscale levels. This gap in understanding restricts the ability to achieve strong and reliable bonding in practical applications. Furthermore, the adhesion strength and stability of hydrogen-bonded polymer adhesives are notably compromised at elevated temperatures, particularly above 100 °C, due to the thermal instability of hydrogen bonds[9,31–33]. In contrast to polymers, oligomers have gained increasing attention for adhesion applications due to their precise structural control, improved processability, and enhanced adhesion performance[13,17,34,35]. However, the development of high-strength and thermal-resistant adhesives based on hydrogen-bonded oligomers remains an unmet challenge. This underscores the necessity for innovative strategies to design and engineer supramolecular oligomeric adhesives capable of sustaining robust adhesion under extreme thermal and mechanical conditions.

This work presents a multiscale engineering approach that integrates molecular, nanoscale, and microscale features into the adhesive matrix. By leveraging controlled solution-shearing processes, we strategically manipulate the formation and orientation of hierarchical nanostructures, including nanoaggregates, nanocrystals, and nanofibrils, driven by hierarchical hydrogen-bonding interactions. This innovative strategy enables simultaneous control over both the adhesion chemistry at the interface and the nano/microstructure of the adhesive layer, achieving a synergistic enhancement in both bulk cohesion and interfacial adhesion across a broad temperature range. This resulting multi-level organization ensures efficient energy dissipation, crack resistance, and chemical and thermal tolerance, allowing the adhesive to maintain its mechanical and adhesion strength even under extreme conditions, where traditional hydrogen-bonded systems typically fail. This research underscores the transformative potential of solution-shearing in designing next-generation adhesives, opening new avenues for applications in demanding environments, such as aerospace, automotive, and electronics.

## Results

### Molecular design and materials characterization

For the design of multiblock oligomers, we leveraged hydrogen-bonded building blocks to self-assemble into semi-crystalline and microphase-separated structures that enhance the bulk mechanical properties of the adhesive. In addition, soft building blocks were incorporated to impart flexibility to the chain segments, facilitating the rearrangement of functional units. Hierarchical hydrogen bonding motifs offer abundant physical crosslinking, imparting mechanical robustness, and enabling robust, reversible adhesion with effective energy dissipation. We synthesized a series of supramolecular thermoplastic oligomers via polycondensation, as shown in Fig. 1a and Table S1. In the molecular design phase, 2-Ureido-4-pyrimidinone (UPy) was chosen for its strong multivalent hydrogen bonding capacity, facilitating robust supramolecular assembly and

polymerization[13,17,36]. Adipic acid units contributed to nanocrystals, while the commercial product Priamine 1074 was selected for its non-crystallizable and soft, flexible properties[17]. These components were combined to form multiblock supramolecular oligomers characterized by semi-crystalline and microphase-segregated structures.

Proton nuclear magnetic resonance ($^1$H NMR) and Fourier transform infrared spectroscopy (FTIR) were performed to characterize the resultant oligomers. As shown in $^1$H NMR spectra (Fig. S1, 2), N-H proton resonances of the UPy units appeared in a downfield region from 10 to 14 ppm, which indicates the assembly and dimerization of supramolecular oligomers assisted by quadruple hydrogen bonding[17,37]. The FTIR analysis indicated the completion of the reaction through the diminished absorbance of the isocyanate group at around 2265 cm$^{-1}$ and the presence of the carbonyl stretching vibration signal at 1632 cm$^{-1}$ (Fig. S3). MALDI-TOF mass spectrometry was performed to determine the molecular weight of the resultant samples. The values shown in Fig. S4 confirm the presence of oligomers. Thermogravimetric analysis was conducted to investigate the thermal stability of the resultant oligomers. Figure S5 indicates that all oligomers exhibited stability across a broad temperature range, with decomposition initiating above 260 °C.

We manipulated molecular packing and nanocrystalline alignment within these oligomers, both in the bulk phase and at the surface layer, through a controlled shearing process (Fig. 1b). By varying the shear rates, we tailored the formation of enlarged, ordered nanocrystals and aligned nanofibrils parallel to the direction of the shear, achieving high and tunable mechanical robustness alongside strong, thermal-tolerant adhesion. Through multiscale engineering, we precisely controlled molecular bonding motifs, nanocrystalline domains, and nanofibrillar architectures to develop a series of high-performance supramolecular adhesives (Fig. 1c). The solution-sheared supramolecular thermoplastic oligomers are denoted as SST-n-m, where n represents the feed quantity of adipic acid, ranging from 1 to 3, and m indicates the applied shear rate, ranging from 0 to 15 mm/s. The oligomer films prepared without solution shearing are designated as SST-n-0.

### Hierarchical nanostructures designed by hydrogen bonding

In this work, the solution-sheared supramolecular oligomers featured the hierarchical nanostructures formed through hydrogen bonding. These nanostructures include nanoaggregates, nanocrystals, and nanofibrils, which play distinct yet complementary roles in enhancing both bulk cohesion and interfacial adhesion: (1) Nanoaggregates are self-assembled clusters of oligomers driven by strong hydrogen-bonding interactions, particularly the quadruple hydrogen bonds formed by UPy units. The nanoaggregates create a network of physical crosslinks within the adhesive, improving energy dissipation and crack resistance. (2) Nanocrystals are ordered, crystalline domains formed by the adipic acid units within the oligomers. The solution-shearing process promotes the growth and alignment of nanocrystals, enhancing the adhesive's mechanical strength and thermal stability. (3) Nanofibrils are elongated, fiber-like structures that are oriented parallel to the direction of shear. These nanofibrils enhance the adhesive's mechanical robustness and toughness by dissipating stress during deformation and providing structural integrity. The synergistic effect of nanoaggregates, nanocrystals, and nanofibrils, stabilized by hierarchical hydrogen bonding, enables the adhesives to achieve unprecedented performance in terms of strength, toughness, and thermal resistance.

### Strong and tough adhesion

The SST-n-0 oligomers can be applied to a variety of substrates following controlled solution-shearing, complete solvent evaporation, and subsequent cooling. We first evaluated the maximal stress and energy required to break the adhesive joints using lap shear strength

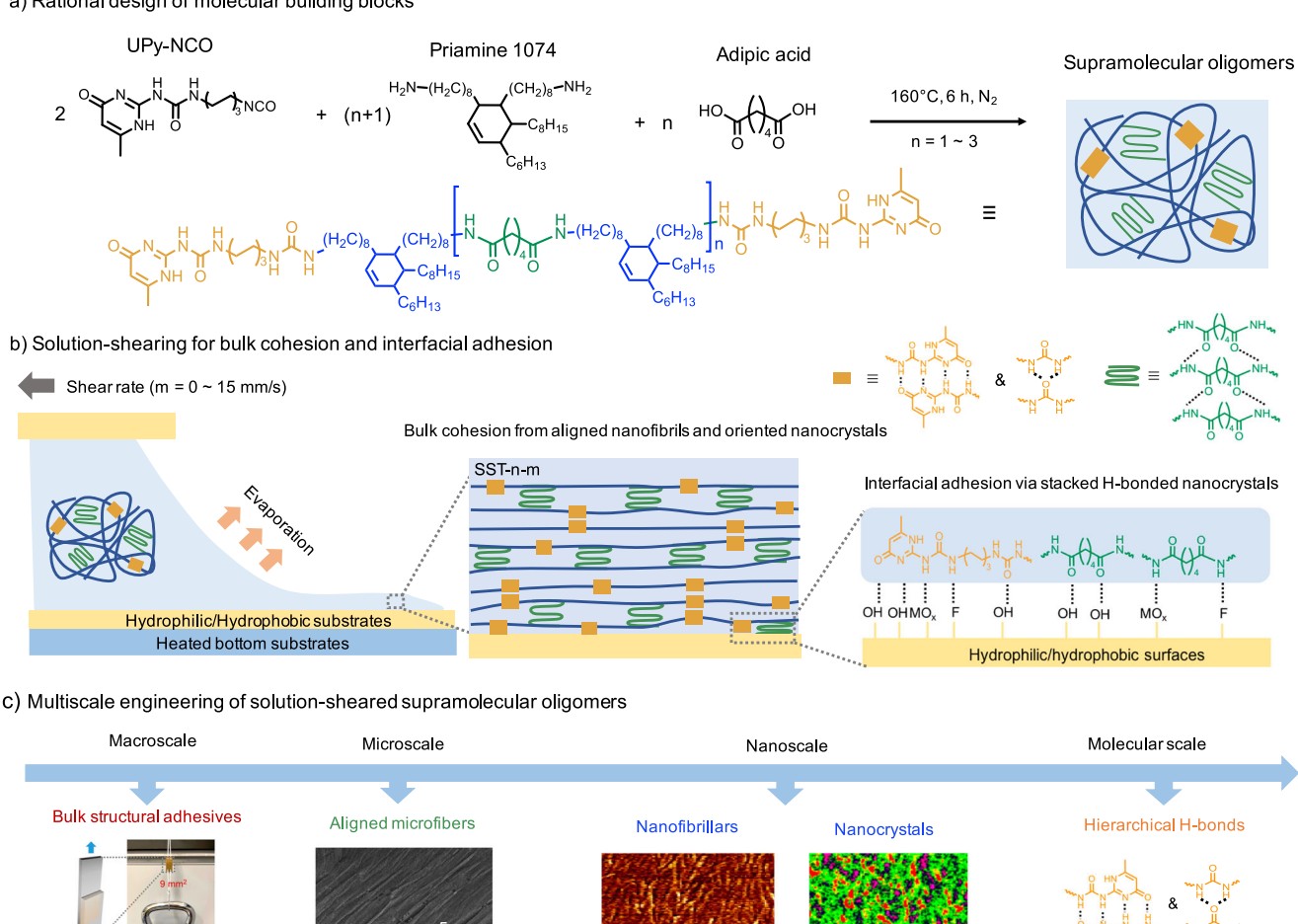

**Fig. 1 | Solution-shearing in hydrogen-bonded supramolecular oligomers.**
**a** Chemical structure and synthesis of supramolecular oligomers. **b** Hierarchical nanostructuring via solution-shearing for enhanced adhesion and cohesion. These ordered nanostructures require higher energy for crack propagation compared to amorphous chains, enabling concurrent optimization of bulk toughness and interfacial bond strength. **c** Multiscale engineering of high-performance adhesives.

and work of debonding measurements in a direction parallel to the nanofibrils (Fig. 2a–c, and S6). Initial tests showed the lap shear strength of unsheared films, including SST-1-0, SST-2-0, and SST-3-0 to be 12.6 ± 0.9 MPa, 23.5 ± 1.0 MPa, and 18.9 ± 1.1 MPa on stainless steel substrates at a pulling rate of 10 mm/min. After the shearing process at 10 mm/s, the lap shear strength of the SST-2-0 on stainless steel increased significantly, reaching 30.6 ± 1.4 MPa. Additionally, the work of debonding rose from 11.7 ± 1.2 kN m⁻¹ to 23.6 ± 1.3 kN m⁻¹. The lap shear stress of SST-2-n was also measured in a direction perpendicular to the nanofibrils. As shown in Fig. S7, no significant difference in lap shear strength was observed between the sheared and unsheared samples.

Further lap shear adhesion testing of SST-2-10 on various substrates (Fig. 2d) in a direction parallel to the nanofibrils revealed distinct performance: on hydrophobic substrates like PTFE (7.1 ± 1.0 MPa) and PMMA (7.8 ± 1.2 MPa), and hydrophilic substrates like epoxy (8.9 ± 1.3 MPa). Notably, on metals and glass, values exceeded 30 MPa, with a peak lap shear stress of 32.2 ± 1.3 MPa on glass slides at room temperature. As a practical demonstration, a SST-2-10 film bonded various substrates together and successfully supported a 10 kg weight for at least 30 days without failure (Fig. 2e). Furthermore, SST-2-10 also showcased exceptional adhesion robustness, reusability, and

recyclability. No significant reduction in lap shear stress was observed after 20 cycles of testing on aluminum substrates (Fig. 2f).

Comparative assessments revealed that the lap shear strength of SST-2-10 on stainless steel is 1.7–6 times greater than that of most commercial hot melt adhesives, specifically being approximately 20 times greater than that of 3 M 2665 polyurethane adhesive (Fig. 2g, Table S4). Furthermore, SST-2-10 markedly outperformed both commercial structural adhesives like Elmer's Glue All and Loctite products, and reported thermoset adhesives across various substrates[38–40] (Fig. 2h, Table S5). For instance, on hydrophilic substrates, such as aluminum, stainless steel, and glass, the lap shear strength of SST-2-10 was 1.6–8 times greater than that of commercial structural adhesives and 1.9–6 times higher than thermoset structural adhesives. On hydrophobic substrates like PTFE, it was 7–14.8 times greater than commercial structural adhesives[38–40]. As indicated in Fig. 2i and Table S6, the sheared adhesives exhibit high lap shear strength and high work of debonding, greatly outperforming the reported state-of-the-art polymer adhesives, including gels[41–43], elastomers[27,44–49], and resins[14,50–54]. These results underscore the superior adhesion performance of solution-sheared adhesives positioning them as a groundbreaking class of high-performance supramolecular structural adhesives.

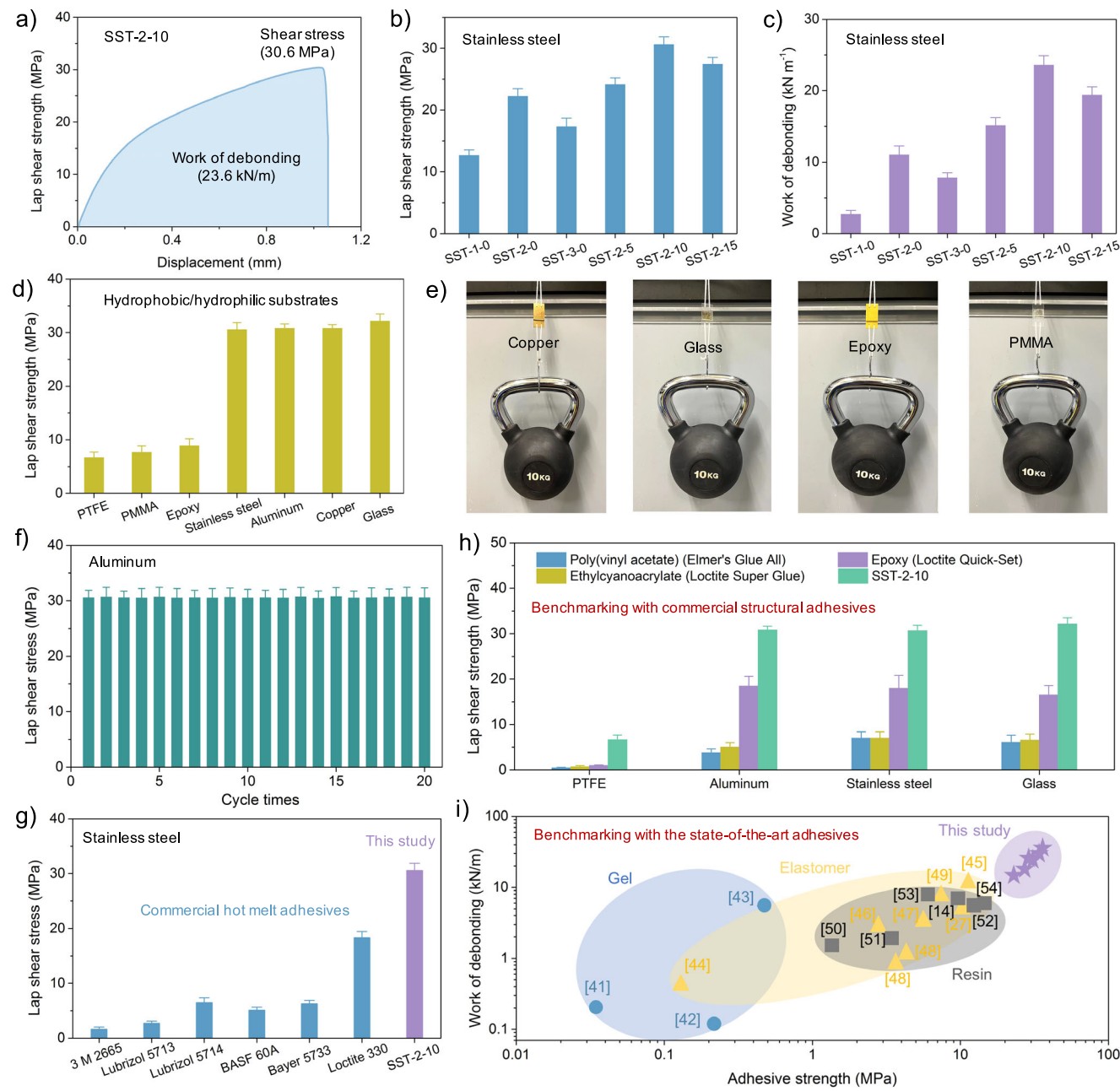

**Fig. 2 | Strong and tough adhesion. a** Lap shear curve of SST-2-10 bonded on stainless steel substrates. Lap shear stress (**b**) and work of debonding **c** of solution-sheared films bonded on stainless steel substrates. **d** Lap shear strength of SST-2-10 bonded on diverse substrates. **e** The demonstration of SST-2-10 maintaining adhesion on substrates after 30 days without failure. **f** Lap shear strength retention of SST-2-10 on aluminum substrates after 20 adhesion cycles. **g** Comparative lap shear strength of SST-2-10 and commercial hot-melt adhesives on stainless steel. **h** Lap shear strength comparison of SST-2-10 and commercial structural adhesives (Elmer's Glue All, Loctite Super Glue, and Loctite Quick-Set) on glass, aluminum, stainless steel, and PTFE. **i** Performance benchmarking of SST-2-10 against reported polymer gels, elastomers, and resin-based adhesives in terms of lap shear strength and debonding energy. All adhesion tests were conducted in the direction parallel to nanofibril alignment. Error bars represent standard deviation (n ≥ 3).

## Chemical-resistant and durable adhesion

To evaluate the chemical resistance and durability of the adhesive, the bonded samples were immersed in various solvents for 24 h. The solvents included: polar solvents (water, saltwater, urea solution, acidic (pH = 1), and basic (pH = 14) solutions), non-polar solvents (hexane), and polar organic solvent (ethanol). The lap shear strength was measured to assess any changes in adhesion performance. The purpose of these tests was to determine stability and durability of the adhesive in environments where it might be exposed to different chemical conditions. This is particularly important for applications in harsh or variable environments, such as industrial, automotive, or biomedical settings. As shown in Fig. S8, the adhesive maintained strong adhesion after exposure to all tested solvents, with only a minor reduction in lap shear strength observed in ethanol (retaining 86% of its original strength). Besides, SST-2-10 can maintain adhesion strength for up to 30 days of continuous water immersion (Fig. S9). The stable adhesion in aqueous conditions is attributed to the robust assembly of hierarchical nanostructures and the shielding of water molecules by the hydrophobic alkyl chains. The resulting adhesives demonstrated excellent resistance to water, acids, bases, and non-polar solvents,

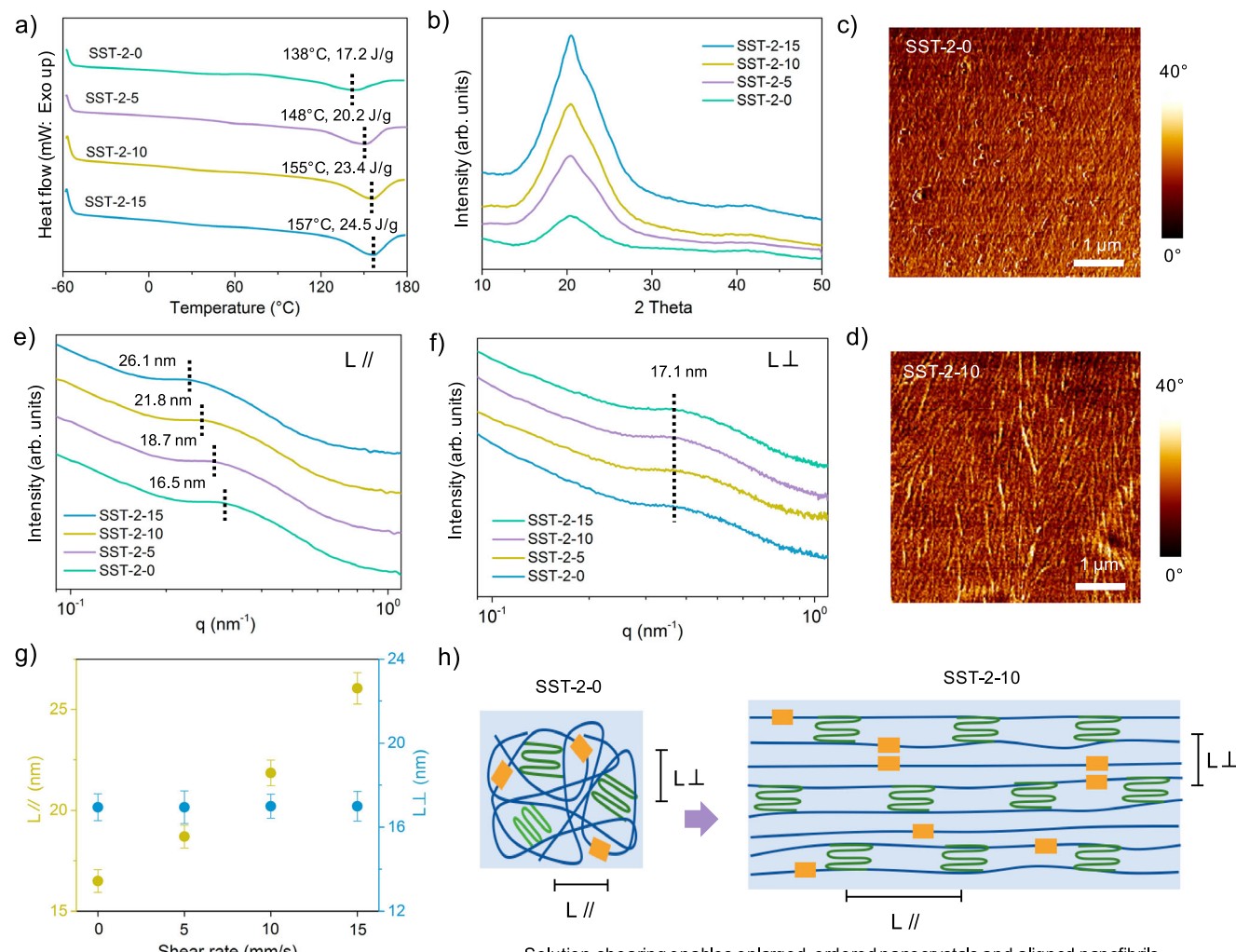

**Fig. 3 | Shearing-induced structural orientations. a** DSC thermograms of SST-2-m films processed at different shear rates (0, 5, 10, and 15 mm/s), recorded during the second heating scan (−60 to 180 °C at 10 °C/min). **b** XRD patterns of SST-2-m on glass substrates processed at varying shear rates (0, 5, 10, and 15 mm/s). AFM phase images showing the nanofibrils arrangement in **c** unsheared (SST-2-0) and **d** sheared (SST-2-10) films on glass substrates. SAXS profiles of sheared films showing scattering intensity versus q vector **e** parallel to (//) and **f** perpendicular to (⊥) the nanofibril alignment direction. **g** The average distance between adjacent nanocrystals parallel (L //) and perpendicular to (L ⊥) the shearing direction for films processed at 0–15 mm/s shear rates. **h** Schematic illustration of the average distance between adjacent nanocrystals before and after shear-alignment. All error bars represent the SD with at least three replicates.

highlighting their robustness and suitability for demanding applications.

## Shearing-induced structural orientations

To elucidate the reasons for high-strength adhesion, the sheared SST-2-m prepared on the glass substrates was chosen, and its structure was investigated by a series of measurements including differential scanning calorimetry (DSC), X-ray diffraction (XRD), small-angle X-ray scattering (SAXS), and atomic force microscopy (AFM). Notably, as the shear rate increased, the glass transition temperatures of the sheared films approached room temperature, while melting temperatures increased from 138 to 157 °C, as determined from the second heating cycle of the DSC curves (Fig. 3a, Table S3). The elevated melting point and increased melting enthalpy correspond to the formation of numerous nanocrystals after solution-shearing. XRD analysis showed that the unsheared adhesives on the glass substrates exhibited a semi-crystalline structure (Fig. S10). With rising shear rates from 0 to 15 mm/s, the intensity of the crystalline peaks increased significantly, and the crystallinity enhanced from 43 ± 2.8 to 60 ± 2.9 % (Fig. 3b, Fig. S11). This enhancement suggests that solution-shearing effectively augments the crystallinity within the oligomers. Further, the crystalline peak of the

SST-2-15 adhesive was deconvoluted, revealing peaks centered at 2 Theta values of 20.18° and 23.76° (Fig. S12), assignable to the (020) and (110) planes of the crystalline methylene groups of the adipic acid block[26,55].

Additionally, the radius of gyration, calculated using the Guinier profile[56], showed significant growth in sizes of nanocrystals from 5.74 to 11.35 nm as the shear rates increased (Fig. S13). The sizes of nanocrystals SST-2-15 were double that of SST-2-0, indicating that shearing markedly promotes the packing of methylene molecules and growth of nanocrystals. AFM phase images provided further insights (Fig. 3c, d). While the unsheared SST-2-0 displayed a random distribution of nanofibrils, the nanofibrils in the sheared SST-2-10 were rearranged and oriented along the shear direction. SAXS analysis complemented these findings, revealing that the average distance between neighboring nanocrystals parallel to the aligned nanofibrils increased from 16.5 to 26.1 nm as the shear rate increased, while the perpendicular distance remained constant (Fig. 3e–h). Furthermore, SST-2-10 exhibited aligned microfibers within the bulk, which was validated by Scanning Electron Microscopy (SEM) image (Fig. S14). The shearing process facilitates the formation of enlarged, ordered nanocrystals and aligned nanofibrils within the oligomers. These high-energy

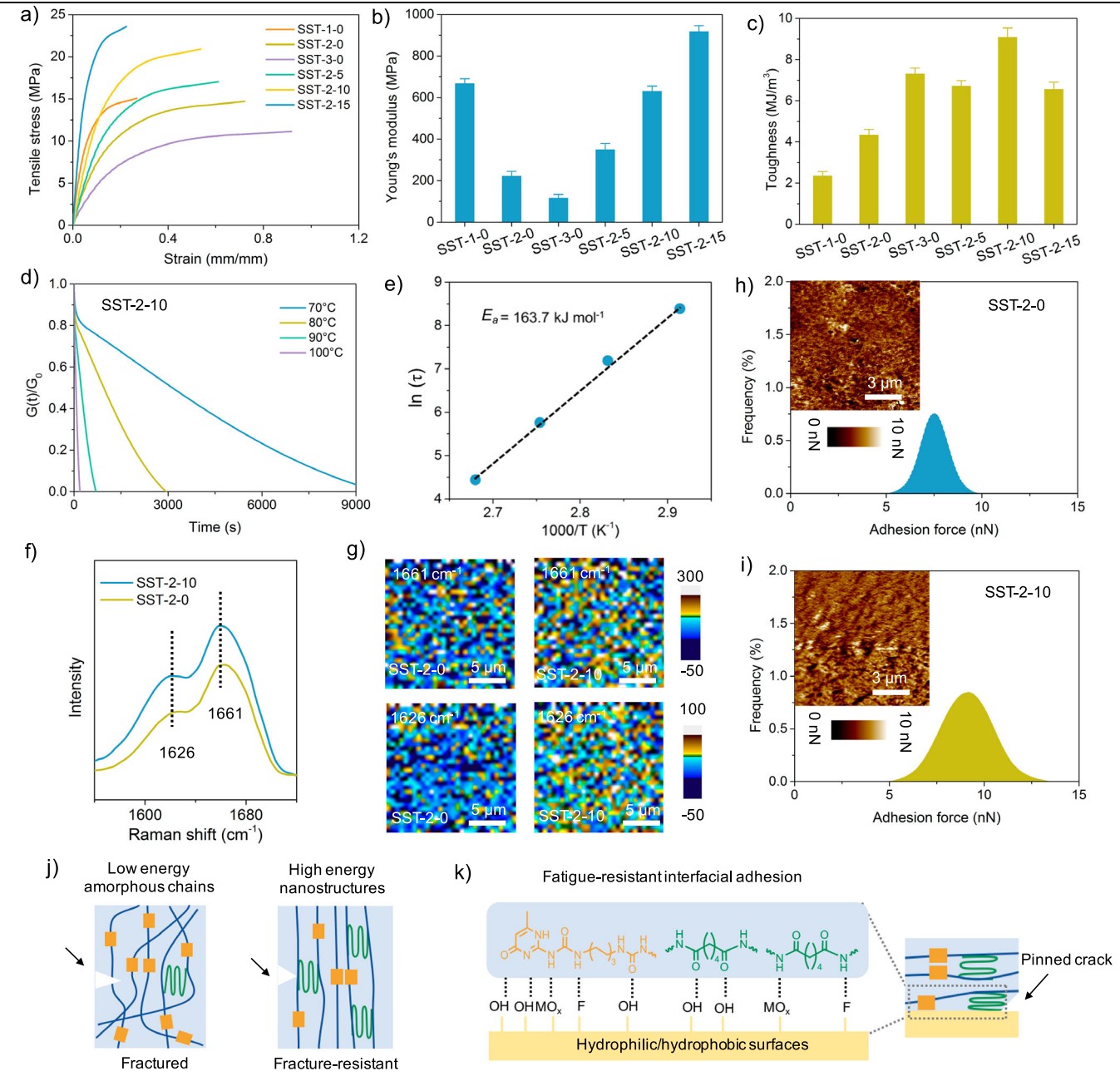

**Fig. 4 | Bulk and interfacial characterizations. a** Uniaxial tensile stress-strain curves of SST-n-m films (stretching rate: 10 mm/min) measured parallel to the nanofibril alignment direction. **b** Young's modulus and **c** toughness of SST-n-m films, derived from tensile testing. **d** Stress relaxation tests of SST-2-10 measured at varying temperatures. **e** The natural logarithm of relaxation times (ln τ) versus inverse temperature (1/T) for SST-2-10, with activation energy ($E_a$) calculated from linear fitting. **f** Raman spectra and **g** surface mapping of dissociated UPy motifs and urea units in SST-2-0 (unsheared) versus SST-2-10 (sheared). Adhesion force distribution maps for **h** SST-2-0 and **i** SST-2-10, demonstrating enhanced interfacial bonding in sheared films. **j** The improved and fracture-resistant cohesion results from ordered nanostructures in the bulk, which effectively resist crack propagation through energy dissipation. **k** Interfacial adhesion reinforcement arises from surface-anchored nanocrystals that pin cracks at the interface. These ordered nanostructures require higher energy for crack propagation compared to amorphous chains, thereby enabling fracture-resistant cohesion and adhesion. All error bars represent the SD with at least three replicates.

ordered nanostructures enhance structural and thermal stability while effectively pinning cracks within the bulk and at the interface. This mechanism enables fracture-resistant bulk cohesion and interfacial adhesion.

## Synergistic enhancement of bulk cohesion and interfacial adhesion

The shearing-enhanced bulk cohesion and interfacial adhesion was systematically investigated (Fig. 4). We first performed tensile stress measurements of free-stranding films in a direction parallel to the nanofibrils. The unsheared SST-n-0 transitioned from stiff, glassy behavior to tough, plastic-like responses as the adipic acid monomer feed increased. Specifically, tensile stress at break decreased from $15.26 \pm 1.15$ to $11.43 \pm 0.95$ MPa, Young's modulus ranged from $667.32 \pm 24.13$ to $116.86 \pm 18.38$ MPa, and tensile toughness increased from $2.35 \pm 0.21$ to $7.42 \pm 0.26$ MJ/m$^3$ (Fig. 4a–c, Table S2). Remarkably, the shearing process significantly enhanced the mechanical strength, modulus, and toughness of the SST films, contributing greatly to bulk

cohesion. For instance, the sheared SST-2-15 exhibited a tensile strength of $23.58 \pm 0.98$ MPa and a Young's modulus of $918.75 \pm 28.28$ MPa, representing 1.6-fold and 4.2-fold increases over the unsheared SST-2-0, respectively. The sheared SST-2-10 film showed an optimal tensile toughness of $9.08 \pm 0.45$ MJ/m$^3$, more than doubling that of its unsheared counterpart. The high mechanical robustness and toughness of the sheared samples stem from the formation of high-energy ordered nanostructures (e.g., nanocrystals, nanofibrils). These nanostructures effectively prevent crack propagation[57], enabling a much higher energy dissipation in contrast to amorphous polymer chains (Fig. 4j). We also conducted the tensile tests in a direction perpendicular to the nanofibrils. The tensile strength of the sheared samples is slightly higher than that of the unsheared SST-2-0 (Fig. S15). However, SST-2-10 exhibits significantly higher fracture strength in the direction parallel to the nanofibrils compared to the perpendicular direction, highlighting the critical role of enlarged, oriented nanocrystals and aligned nanofibrils in enhancing mechanical robustness and toughness along the fibril alignment axis.

The dynamic mechanical behavior of the SST-2-0 and SST-2-10 was explored through stress relaxation experiments at various temperatures (Fig. 4d, e, and S16). At a constant strain of 1%, stress decay and relaxation rates increased with temperature (Fig. 4d). SST-2-10 exhibited distinct stress relaxation behaviors at different temperatures. At low temperatures, the topological network of SST-2-10 is effectively frozen, resulting in slow stress relaxation. In contrast, at elevated temperatures, the relaxation process is significantly accelerated due to the increased mobility of the supramolecular network. This fast response time is attributed to the reversible nature of the hydrogen bonding interactions within the supramolecular network, which allows the material to rapidly reorganize its structure in response to thermal stimuli. Furthermore, the hierarchical nanostructures of SST-2-10, comprising nanocrystals and aligned nanofibrils, facilitate efficient energy dissipation and structural reorganization, contributing to their enhanced thermal responsiveness. The characteristic relaxation time ($\tau$), calculated at 1/e (37%) of the normalized relaxation modulus, was over three times lower for SST-2-10 at 70 °C compared to SST-2-0, indicating restricted chain mobility and bond dissociation. The apparent activation energy for stress relaxation, calculated by fitting the relaxation time versus temperature[58,59], was 163.7 kJ/mol for SST-2-10 (Fig. 4e), which was nearly 1.6 times higher than that of SST-2-0 (Fig. S16). This high activation energy, superior to most reported covalently crosslinked and supramolecular polymers[27,59,60], In addition, dynamic physical crosslinking within the adhesives facilitated easy reprocessing and reuse. SST-2-10 witnessed no significant changes in mechanical properties after three-time tensile processes, demonstrating the material's potential as reusable and recyclable structural materials (Fig. S17).

We further conducted the interfacial characterization to explore chemical components, modulus, and adhesion force of SST film surface layers. As analyzed in Raman spectra and mapping, the surface layer of SST-2-10 showed higher intensities of dissociated UPy motifs at 1661 cm$^{-1}$ and ordered urea units at 1626 cm$^{-1}$ compared to SST-2-0 (Fig. 4f, g). To further confirm the presence of hydrogen bonding at the surface, ATR-FTIR analysis was conducted on the adhesives surface. As shown in Fig. S18, upon heating from 30 to 130 °C, the C = O stretching band gradually shifts from 1654 to 1670 cm$^{-1}$, while the N-H band shifts from 3328 to 3350 cm$^{-1}$. These shifts reveal a weakening of hydrogen bonding interactions at elevated temperature, confirming the presence of the hydrogen bonding units that can be reconfigurable during the solution shearing process. We also investigated the surface stiffness or elasticity of the adhesive coatings using Peak Force Quantitative Nanomechanical Mapping (Fig. S19). The surface modulus was probed with a sharp tip and the force-displacement response was analyzed. This analysis provides valuable information about the mechanical properties of the adhesive surface, which are critical for

understanding its performance in practical applications. In contrast to the surface of SST-2-0, SST-2-10 had a higher surface modulus (Fig. S19), thus suggesting more nanocrystalline domains accumulated at the SST-2-10 surface, facilitated by the solution-shearing process. AFM indentation measurements (scan area: $9 \, \mu m \times 9 \, \mu m$, testing points: $256 \times 256$) further elucidated the adhesion force distribution and mapping of the surface layers (Fig. 4h, i), with SST-2-10 exhibiting an average adhesion force of 9.1 nN, greater than SST-2-0. The enhanced interfacial adhesion of SST-2-10 aligns with ordered nanostructures that effectively pin cracks at the interface and within the bulk[24]. Besides, with solution shearing and cooling process, the surface chemical composition of the adhesive is effectively reconfigured (Fig. 4f, g) and abundant nanocrystals are anchored to the substrate (Fig. S19), which is beneficial for fatigue-resistant interfacial adhesion (Fig. 4k). The high density of hydrogen bonding units, e.g., dissociated UPy motifs, urea groups, and amide units, at the surface of sheared adhesives, can strongly interact with functional sites like hydroxyl units of glass, oxygen units of metallic oxides, or fluorine of PTFE, at the substrates. These findings underscore the interplay between ordered nanocrystals and abundant hydrogen bonds at the interface, significantly promoting interfacial adhesion in the sheared adhesive coatings.

To further clarify how the solution-shearing process synergistically enhances both bulk cohesion and interfacial adhesion, we quantified interfacial fracture toughness and fatigue threshold via 90° peel testing on stainless steel substrates under a single cycle and multiple cycles of loads[24,57,61]. As indicated in Fig. S20, SST-2-10 yielded an interfacial fracture toughness value of 6700 J/m$^2$, 3.7-fold higher than SST-2-0 (1800 J/m$^2$), which confirms that solution-shearing enhances interfacial adhesion through hydrogen-bonded nanocrystals anchoring. Further, cyclic peeling tests revealed an interfacial fatigue threshold of 1360 J/m$^2$ for SST-2-10, compared to 230 J/m$^2$ for SST-2-0, demonstrating that aligned nanofibrils in the bulk phase resist crack propagation and effectively dissipate energy during repeated loading. Together, these decoupled metrics validate the dual-control mechanism underpinning our synergistic design.

## Thermal-resistant adhesion

Figure 5a reveals that SST oligomers maintain stable adhesion at elevated temperature through temperature-adaptive nanostructures: high crystallinity (>50% at 90 °C) provides mechanical integrity, while dynamic hydrogen bonding networks enable stress relaxation above 100 °C. Both features demonstrate the ability of SST oligomers to maintain mechanical performance through gradual thermal transitions, in contrast to conventional adhesives that suffer abrupt property degradation near their melting points. Lap shear adhesion measurements on copper substrates bonded with SST-2-10 demonstrated consistent adhesion strength exceeding 20 MPa across a broad temperature range from 20 to 120 °C. Beyond 120 °C, the adhesion strength gradually decreased. These findings are consistent with the observed changes in crystallinity at elevated temperatures (Fig. S21). At 90 °C, the crystallinity remained above 50%, resulting in a lap shear adhesion strength exceeding 25 MPa. At 130 °C, with crystallinity above 20%, the adhesion strength remained above 15 MPa. At 150 °C, where crystallinity dropped to around 10%, the adhesion strength decreased to 5.3 MPa. We noted that the highest adhesion strength of SST-2-10 was 33.8 MPa at 70 °C (Fig. 5b). This optimal performance is attributed to the rubbery state of the UPy-functionalized crosslinks at this temperature, which enhance network flexibility and toughness[32]. This state allows the material to withstand maximum formation before failure, thereby maintaining structural integrity and increasing adhesion strength. Additionally, the lap shear strength of SST-2-10 did not significantly decrease after 10 cycles at 50 °C and 150 °C, respectively (Fig. 5c). We hypothesized that the thermal-resistant adhesion properties of SST-2-10 is due to hierarchical hydrogen-bonded

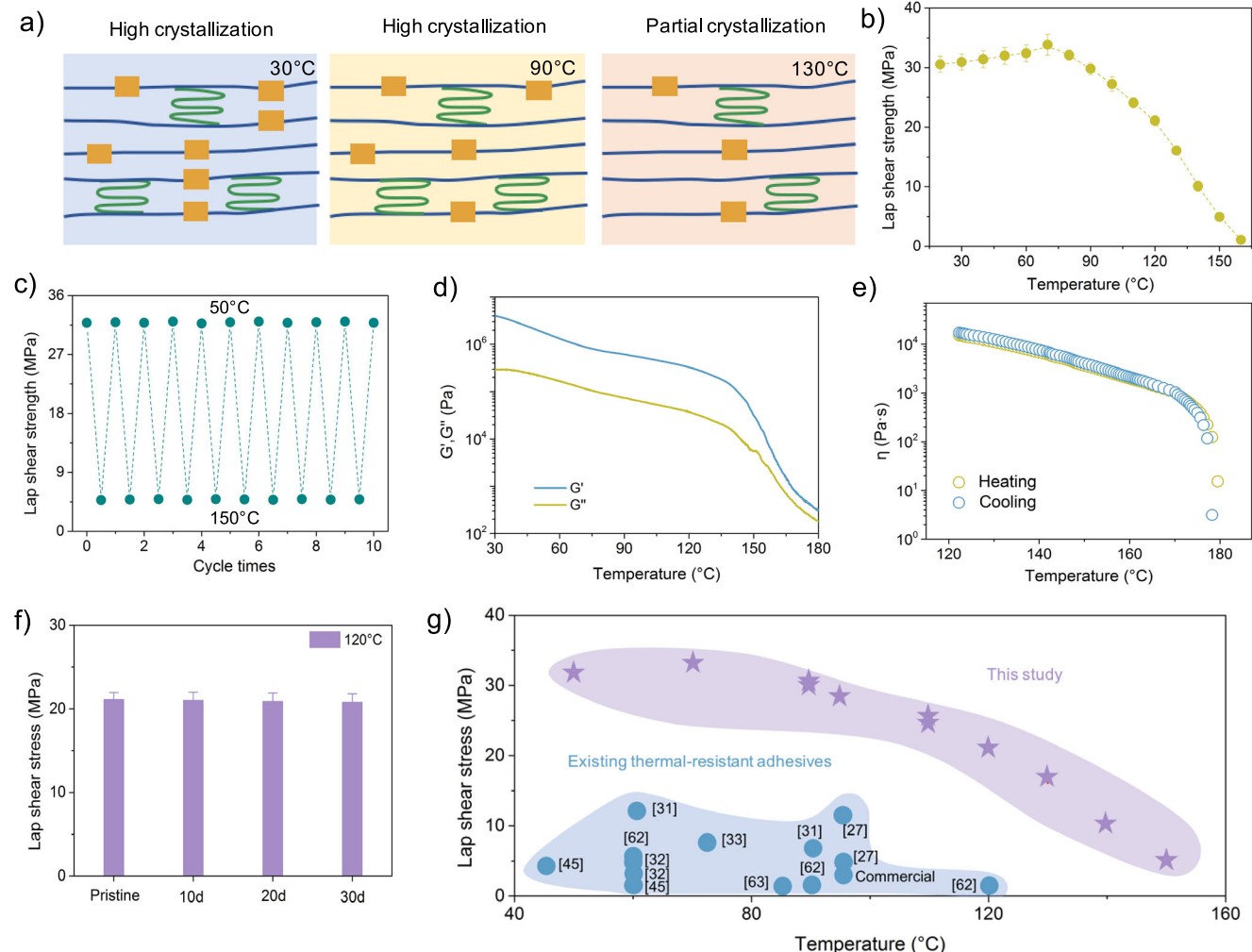

**Fig. 5 | Thermal-resistant adhesion. a** Evolution of hierarchical nanocrystalline domains in SST-2-10 under thermal treatment. **b** Lap shear strength of SST-2-10 tested at ambient temperature ranging from 20 to 160 °C. **c** Cyclic lap shear strength retention of SST-2-10 on copper substrates after 10 adhesion cycles at 50 °C and 150 °C. **d** Temperature-dependent viscoelastic behavior of SST-2-10 at a constant frequency of 1.0 rad/s from 30 to 180 °C. **e** Temperature-dependent viscosity (η) of SST-2-10 during heating and cooling cycles (120–180 °C), demonstrating thermal reversibility. **f** Long-term lap shear strength stability of SST-2-10 at 120 °C over 10–30 days. **g** Comparative lap shear strength of SST-2-10 versus reported adhesives across a broad temperature range. All error bars represent the SD with at least three replicates.

nanostructures. To verify this, we first analyzed the temperature-dependent changes in modulus and viscosity. As indicated in Fig. 5d, the storage modulus exceeds the loss modulus at temperatures up to 180 °C, indicating a solid-like behavior of SST-2-10. The modulus analysis also revealed a gradual decrease from 20 to 130 °C, followed by a sharp decline between 130 and 180 °C, coinciding with the phase transition of hierarchical nanostructures within SST-2-10. The viscosity measurements showed that SST-2-10 maintained a high viscosity over 10 k Pa·s at 130 °C, with a gentle decrease as temperature rose to 160 °C before dropping to 10 Pa·s at 180 °C (Fig. 5e). The high viscosity at temperatures below 130 °C featured a high resistance to flow, attributed to the ordered nanocrystals and aligned nanofibrils, enhancing the material's mechanical robustness and thermal tolerance. The lower viscosity above 180 °C is induced by the phase transitions of hierarchical nanostructures, which enables easy handling and processability. The solution-shearing-engineered hierarchy—combining (i) interfacial nanocrystal anchors, (ii) bulk-aligned nanofibrils, and (iii) dynamic bond reconfiguration—explains the exceptional retention of 21 MPa adhesion at 120 °C, where most polymer adhesives experience dramatic performance degradation.

For the long-term adhesion performance, SST-2-10 bonded to glass substrates was held at a sustained temperature of 120 °C for up to 30 days and was tested at given times. The lap shear strength exceeded 20 MPa throughout the testing period, indicating exceptional thermal stability and durability (Fig. 5f). When compared with literature-reported adhesives[27,31–33,45,62,63], SST-2-10 demonstrated superior thermal-resistant adhesion across a wide temperature range. For instance, at 90 °C on glass substrates, SST-2-10 exhibited a lap shear strength of 30.4 MPa, higher than the 6.5 MPa of the reported hydrogen-bonded adhesive P(nBuA-co-Ba-co-HW)[31] (Fig. 5g and Table S7). Additionally, at 95 °C on stainless steel, SST-2-10 adhesion strength of 28.7 MPa greatly outperformed the 11.4 MPa of the dynamic boronic covalent adhesive[27]. These findings underscore the significance of ordered nanocrystals and aligned nanofibrils in SST-2-10, positioning it as a high-performance option for applications in demanding thermal environments.

## Discussion

High-temperature-resistant adhesives and hydrogen-bonding mechanisms have been extensively studied, however, our work

introduces several unique advancements that distinguish it from prior reports: (1) Solution-Shearing Process. Unlike traditional approaches, our solution-shearing process represents a novel approach to simultaneously control both the interfacial adhesion chemistry and the nano/microstructure of the adhesive layer. This dual control enables a synergistic enhancement in bulk cohesion and interfacial adhesion, which has not been achieved in prior studies. (2) Hierarchical Nanostructures. The formation of oriented, enlarged nanocrystals and aligned nanofibrils through solution-shearing is a structural breakthrough not observed in previous adhesive research. These high-energy nanostructures facilitate thermal-resistant interfacial adhesion and bulk cohesion. (3) Advancements in Performance. By leveraging solution-shearing, we have established a new benchmark for thermal-resistant hydrogen-bonded adhesives, offering a versatile approach to optimizing mechanical, thermal, and adhesive properties. Our findings not only reinforce the importance of the solution-shearing process but also provide a novel framework for designing next-generation adhesive materials with tailored properties for demanding applications.

While optimized epoxy adhesives can exhibit higher lap shear strength (10–40 MPa) under ideal conditions and surface treatment[14,64], the SST platform represents a significant shift: offering competitive adhesion on untreated surfaces, combined with thermal stability, reusability, and reversible adhesion—functionalities unattainable in conventional epoxy systems. These features render SST adhesives particularly suitable for emerging applications requiring repeated bonding, thermal resilience, and sustainable material use.

In summary, this study introduces a family of hydrogen-bonded supramolecular oligomers that achieve a synergistic enhancement of thermal-resistant bulk cohesion and interfacial adhesion through a controllable solution-shearing process. This innovative approach enables mechanical robustness, toughness, and strong adhesion to both hydrophilic and hydrophobic substrates across a broad temperature range. Notably, SST-2-10 exhibits a lap shear strength exceeding 30 MPa on metal and glass substrates at room temperature, surpassing conventional commercial hot-melt and structural adhesives. The remarkable performance stems from hierarchical ordered nanostructures (e.g., nanoaggregates, nanocrystals, and nanofibrils) within the bulk and anchored at the surface, synergistically boosting bulk cohesion and interfacial adhesion. Benefiting from the stable and abundant nanocrystalline domains, SST-2-10 maintains adhesion strength over 21 MPa at 120 °C, outperforming current leading commercial options. This research highlights the transformative potential of solution-shearing in designing next-generation adhesives with unprecedented thermal resistance, mechanical strength, and durability. By simultaneously controlling interfacial adhesion chemistry and bulk micro/nanostructure, it opens new avenues for aerospace, automotive, and electronics applications. This work sets a new benchmark for thermal-resistant hydrogen-bonded adhesives, paving the way for sustainable, high-performance materials in demanding environments.

Though the present study focuses on fundamental material mechanisms using lab-scale shear coating, the SST oligomer system is intrinsically compatible with scalable processing. The combination of low viscosity, minimal entanglement, rapid nanostructure formation, and melt-processability positions it well for integration into roll-to-roll or slot-die coating workflows. Future work will quantify large-area performance, leveraging these insights to bridge the gap between laboratory innovation and industrial deployment.

## Methods

### Materials
Dimer diamines (Priamine 1074) were purchased from CRODA Coatings & Polymers. 2-Amino-4-hydroxy-6-methylpyrimidine (99%) was purchased from Acros Organics. The rest of the chemicals and solvents were obtained from Sigma-Aldrich and used exactly as received.

### Synthesis of 2-(6-isocyanato-hexylamino)-6-methyl-4[1H]-pyrimidinone (UPy-NCO)
2-(6-Isocyanato-hexylamino)-6-methyl-4[1H]-pyrimidinone (UPy-NCO) was prepared as reported[17,36]. 2-Amino-4-hydroxy-6-methylpyrimidine (10 g, 79.9 mmol) was added to a 250 mL round bottomed flask. Hexamethylene diisocyanate (HMDI, 100 mL, 624 mmol) and pyridine (7 mL) were then added, the flask was fitted with a reflux condenser, and the mixture was stirred at 100 °C overnight under dry nitrogen. Pentane (30 mL) was then added and the solid product was collected by filtration. The white powder was washed three times with 125 mL portions of acetone to remove unreacted HMDI and then dried overnight under high vacuum conditions at 60 °C (yield 95%).

### Synthesis of supramolecular thermoplastics
UPy-NCO (1.758 g, 2 mmol), adipic acid with various molar ratios and DMSO (45 mL) were added into a 500-mL three necked flask equipped with a reflux cooler. Table S1 details the weight and molar quantities of reactants utilized in this study. The reactor was heated to 160 °C under 300 rpm mechanical stirring and flushed with a continuous nitrogen flow. After complete dissolving of solids and mixing well of the mixture, Priamine 1074 with corresponding content that was pre-dissolved in chloroform (5 mL) was added dropwise. After 6 h, the reaction was checked by infrared spectroscopy, where the absorbance of the isocyanate group at around 2265 cm$^{-1}$ and the carboxylate unit at 1400 cm$^{-1}$ were found to be diminished and the carbonyl stretching vibration signal at 1632 cm$^{-1}$ was present in the final product, indicating the reaction was completed. The solvent was removed, and the solid product was washed three times with 60 mL portions of acetone. The resultant products were collected by centrifugation and subsequently dried overnight under high vacuum at 60 °C, respectively.

### Preparation of solution-sheared supramolecular thermoplastics (SST)
The SST solution was prepared in DMSO at a concentration of 10 mg/mL and then heated in a glass dish at 160 °C for 30 min to evaporate the majority of the solvent. The transparent viscous SST was dropcasted on a pre-heated substrate plate held at 160 °C, while the shearing plate was placed at a tilt angle of 1° from the horizontal level. Both the substrate plate and the shearing pleate were held in vacuum. The gap distance between the substrate plate and the shearing plate was fixed at 200 ± 10 μm. The thickness of the adhesive films can be kept consistent accordingly. The shearing plate was applied and moved along a certain direction with controlled shear rates ranging from 0 mm/s to 15 mm/s, respectively. The resulting sheared film between the substrates was left on the heating plate at 60 °C for 12 h to ensure the complete removal of any residual DMSO. The absence of residual DMSO was confirmed by FTIR and TGA analysis (Figs. S22, 23). After cooling to room temperature, the resulting sheared films can be used for adhesion measurements or peeled off from the substrates for characterizations.

### General characterization
Molecular weight of the monomer was determined by MALDI–TOF mass spectrometry using a Bruker model Autoflex TM speed spectrometer. FT-IR spectra were recorded at room temperature using a Fourier transform infrared spectrometer (PE Spectrum 100) in the range from 4000 to 400 cm$^{-1}$ at a resolution of 4 cm$^{-1}$ and with an accumulation of 16 scans.

### Thermal characterization
TGA was conducted using a thermal analysis (TA) Q600 differential thermal analyzer (DTG) in N$_2$ at temperatures ranging from 30 to 600 °C with a heating rate of 10 °C/min and a sample of 10 mg. DSC measurements were carried out in N$_2$ using a MDSC 2910 system with a heating/cooling rate of 10 °C/min in the temperature range −60 to

180 °C and a sample of 5 mg. Data from the second heating cycle and the reverse heat flow curve are presented unless otherwise indicated.

## Structural characterization
The X-ray diffraction (XRD) analysis (Bruker AXS, D2 PHASER) was conducted with Cu Kα radiation (λ = 0.15418 nm) operating at 30 kV. Phase separation was determined by Small Angle X-ray Scattering (SAXS) measurements that were carried out on SAXSpace[10,65]. The AFM phase images of SST were conducted with a tapping mode on Bruker Dimension Icon with ScanAsyst. The modulus tests were conducted in Peak Force Quantitative Nanomechanical Mapping Method[10,66,67]. Adhesion force distribution depicted in Fig. 4h and i was fitted using a Gaussian function.

## Mechanical characterization
Rheological measurements were performed with a Malvern Kinexus Lab+ in oscillatory mode and a 20-mm diameter parallel plate. A 1 mm gap was included in the setup for rheological testing. Temperature sweep experiments were conducted at a constant strain of 0.1% and a constant frequency of 10 rad/s at temperatures ranging from 30 to 180 °C. Mechanical tensile-stress testing was carried out at room temperature using MTS Alliance RT/30 tensile tester with a sample size of 30 mm length × 6 mm width × 3 mm height, a gauge length of 10 mm, and a strain rate of 10 mm/min. Tensile Young's modulus was determined from the slopes of the linear region in the strain regime of 0–0.5%. The area beneath the stress-strain curve was used to calculate tensile toughness. All error bars are standard deviations, and the reported average data comes from three separate tests.

## Temperature-dependent viscosity measurement
The temperature-dependent viscosity tests were performed at temperatures ranging from 120 to 180 °C with a heating/cooling rate of 5 °C/min at a constant strain of 0.1% and a constant frequency of 1 rad/s.

## Stress relaxation measurement
Stress relaxation testing on an oligomeric film with dimensions of 1 mm thickness and 20 mm diameter was performed on a rotational rheometer (Malvern Kinexus Lab+) at 1% strain using a rotation mode. The temperature dependence of the relaxation time was fitted by the Arrhenius Eq. (1)[58,59]:

$$\tau(T) = \tau_0 \exp\left(\frac{E_a}{RT}\right) \tag{1}$$

Where $\tau(T)$ is the relaxation time, $E_a$ is the activation energy, and T is the temperature.

Take the natural logarithm of Eq. (2):

$$\ln[\tau(T)] = \ln\tau - \frac{E_a}{RT} \tag{2}$$

Thus, $E_a$ can be obtained from the slope of ln [$\tau$ (T)] -1/T curve.

## Adhesion measurement
The substrates were sonicated with soapy water, acetone, ethanol, and deionized water for 30 min each, and were then air-dried overnight before being used. The single-lap shear strength tests were conducted using INSTRON-5566 based on the ASTM D1002 standard at room temperature with a strain rate of 10 mm/min on a MTS Alliance RT/30 tensile tester. The thickness of the substrates used was 2 mm. The substrates were sandwiched with a sheared film thickness of ca. 200 μm and a bonded area of 9 mm × 9 mm, namely 81 mm². Lap shear stress was tested in a direction parallel or perpendicular to the oriented nanofibrils. Lap shear strength is defined as the greatest force divided by the overlap area of the sheared film[27]. The integrated area under the force-displacement curve was used to calculate the work of debonding, namely adhesion energy[27,68]. For the cycle tests, the adhesive left on the substrates was collected after the initial adhesion test. The residual adhesive was removed by DMSO. This step is crucial because it ensures that any uneven or degraded areas from the first test do not interfere with the subsequent bonding. The collected adhesive sheets were re-dissolved in DMSO to prepare a 10 mg/mL solution, and we prepared the re-oriented films on the substrates, using the solution shearing method. We then monitored the adhesion strength during each cycle. The standard deviation is cited with the reported shear strength values, which are the averages of three samples.

## Thermal-resistant adhesion measurement
High-temperature lap shear tests were performed isothermally at each target temperature (20 °C–160 °C). Bonded samples were pre-equilibrated in an environmental chamber for 10 min to reach thermal equilibrium before mechanical testing. Shear testing was carried out directly at the test temperature using an INSTRON-5566 equipped with a temperature-controlled chamber, with a constant loading rate of 10 mm/min in accordance with ASTM D1002. The substrates, with a thickness of 2 mm, were bonded using a sheared adhesive film with a thickness of approximately 200 μm and a bonded area of 9 × 9 mm. The lap shear stress was measured in a direction parallel to the oriented nanofibrils. Lap shear strength was calculated as the maximum force divided by the overlap area of the adhesive film. Reported shear strength values represent the average of three samples, with standard deviations provided for statistical reliability.

## Solvent resistance measurement
The sheared films bonded with glass sheets were soaked in diverse solvent, such as water, saltwater, urea, acidic (pH = 1) and basic (pH = 14) suffer, hexane, and ethanol. The adhesive joints were immediately evaluated for lap shear strength after soaking for 24 h.

## Data availability
The data supporting the findings of this study are available within the Article and its Supplementary Information. All data are available from the corresponding author upon request. Source data are provided with this paper.

## Code availability
The simulation codes used in this study are available from the corresponding authors on request.

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

## Acknowledgements

This work was supported by the EPSRC Program Manufacture Using Advanced Powder Processes (MAPP) EP/P006566 (E.S.), the National Natural Science Foundation of China (22305230) (D.W.), Fundamental Research Funds for the Central Universities (21623334) (W.S.), and Guangdong Basic and Applied Basic Research Foundation (2023A1515110825) (W.S.).

## Author contributions

G.L. led this work, including conceptualization, experimental design, characterizations, data interpretation, and manuscript preparation. G.L. and M.L. discussed and analyzed the results. R.M., Y.Z., D.W., W.S., S.A.K., and H.C. assisted with experiments. E.S. provided scientific guidance and valuable suggestions. G.L., E.S., and M.L. revised the manuscript. G.L., M.L., and D.W. supervised the study. All coauthors read and commented on the manuscript.

## Competing interests

The authors declare no competing interests.
