## [Transparent Peer Review file · Nature Communications]

Solution-Sheared Supramolecular Oligomers with Enhanced Thermal Resistance in Interfacial Adhesion and Bulk Cohesion

Corresponding Author: Dr Gang Lu

Version 0:

Reviewer comments:

Reviewer #1

(Remarks to the Author)

Please specify the chemical structure of the entire polymer at the early part of the manuscript. Also, include schemes that illustrate the polymer synthesis process. These should be presented in or before Figure 1 in the main text rather than in the supporting information.

The explanation of the multiscale adhesion mechanism looks good; however, are these demonstrated mechanisms hypotheses or proven scientific concepts? The detail discussion / explanation is required.

On line 215, what do 020 and 110 in parentheses represent? Please specify these numbers.

What is the ratio among the three segments in the polymer?

If you determined this ratio, then why did you pick this particular ratio? Did you try other ratio? If not why not?

Are the polymer segments randomly aligned, block, or exhibit any regularity? To avoid such fundamental questions, please discuss the polymer synthesis in detail.

The molecular weight information of the polymer should be specified in the main text. Any GPC (SEC) data?

Is there any crystallinity change over time? If the crystallinity changes after a few days of storage, the polymer's overall properties could also change. Additionally, crystallinity can change at different temperature ranges. Test results or, at the very least, a well-thought-out explanation is required.

The test method regarding solvent use and influence is somewhat confusing. Adhesion was tested after the complete drying of the solvent; what, then, is the purpose of using multiple solvents? To enhance clarity, the author should separate the solvent effect section. The current writing in the manuscript may cause confusion.

Please correct the typo in Figure S16's caption.

In Figure S17, the term "Surface modulus on AFM" should be accompanied by brief background information.

Table S7 needs lines to specify the source.

What are the key reasons for strong adhesion at high temperatures?

The term "Ultra-high" should be removed from the title.

The author should provide an explanation of the "solution shared process." This concept is crucial to the paper but is not explained.

The concepts of H bonding in "nano aggregates" and "nanocrystals" must be thoroughly explained. These terms are used frequently in the manuscript without clear definitions.

How are "Hierarchical aggregates" formed? Please provide a clear definition.

On line 111, adipic acid contains an aliphatic linkage between functional groups, meaning individual segments do not exhibit rigidity. Please delete the term "rigid." While adipic acid can form a crystallizable block, individual adipic segment does not confer rigidity.

Please define "ST polymer" and "SST" at least once in the manuscript before using these acronyms.

Can this work be applied in real-world scenarios? The presented control parameters, such as solvent evaporation, temperature control, and shearing control, appear too delicate for practical applications.

The influence of the applied shear rate on adhesion properties is well-established textbook knowledge and has been reported extensively in adhesion and polymer science for decades. Reporting this effect may be redundant.

This manuscript contains extensive benchmarking tests against commercial adhesives and previously reported adhesives. However, the main scientific or engineering discovery from this study is unclear. The manuscript appears more suitable for an engineering and/or application-focused journal.

While the manuscript presents high-temperature-resistant adhesives, numerous reports already exist on such materials. Moreover, the mechanism based on hydrogen bonding from known functionalities (e.g., Upy) is well-documented. The author uses vague terms but fancy terms such as "solution sharing process" and "Hierarchical aggregates," which are not clearly defined in the manuscript. These terms might mislead editors and readers by describing well-known textbook knowledge with new and unclear terminology. I have serious concerns regarding the novelty of this manuscript.

Reviewer #2

(Remarks to the Author)

Designing strong and thermally resistant adhesives has become a key research focus for load-bearing applications. The authors present a strategy involving solution-sheared supramolecular oligomers, which exhibit excellent adhesive strength and toughness across a broad temperature range. This adhesive demonstrates impressive lap shear strength, surpassing that of commercial structural and thermoset adhesives. However, several important issues need to be properly addressed before the manuscript can be considered for publication.

1. Why did the authors choose the solution-sheared supramolecular oligomer strategy? Why opt for an oligomer instead of a polymer?
2. The quality of the ¹H NMR spectrum data for ST-2 is low. Some proton peaks remain unassigned.
3. The authors describe SST as a polymer film. However, this contradicts the MALDI-TOF mass spectrometry results, which suggest it is an oligomer with low molecular weight.
4. Was the failure mode during adhesion tests cohesive failure, adhesive interfacial failure, or a combination of both? Proper experimental evidence and discussion should be provided. Additionally, SST-2-10 exhibits outstanding lap shear strength, surpassing that of many commercial adhesives. The authors should conduct a cost analysis to assess its feasibility as a next-generation adhesive material.
5. The adhesion mechanisms of the developed adhesive on various substrates should be more thoroughly characterized and discussed.
6. In Figures 2i and 5e, literature sources should be cited for performance comparisons with previous works. Additionally, detailed testing conditions for the reported thermal-resistant adhesives should be provided.
7. An AFM topography image of SST-2-10 should be included to further support its aligned nanofibril structure.
8. The authors should verify the frequency values in the AFM adhesion force measurements and consider providing fitted Gaussian distribution curves.
9. The response time of SST-2-10's mechanical properties to temperature variations should be discussed in Figure 5.

Reviewer #3

(Remarks to the Author)

This work describes a class of solution-sheared supramolecular oligomers with strong adhesive strength and toughness across a broad temperature range. However, this work is a moderate extension of previous reported adhesives composed of Upy and/or urea units, there is not much novelty within. This mechanism explanation is not clear enough, the obtained data is insufficient and superficial. There is also no cutting-edge findings or striking progress towards the field of H-bonded adhesives. Therefore, this results appear to be more suitable for publication in a journal like Polymer Journal, does not appear to be of enough urgency for high-ranking Nature Communications.

Moreover, there is a significant issue need to be clarified, in line 388-399, the authors mention that "Solution-shearing of supramolecular thermoplastics (SST). The SST solution was prepared in DMSO at a concentration of 10 mg/mL and then was heated in a glass dish at 160°C for 30 min. left on the heating plate at 60°C for 12 hours until the residual solvent

was completely removed."We know that the boiling point of DMSO is as high as 189 oC, this reviewer is wondering how does the residual DMSO could be completely removed by heating 160 or 60 oC, the evidence regarding the complete removing of DMSO must be provided.

Version 1:

Reviewer comments:

Reviewer #1

(Remarks to the Author)

[Editor Note: This reviewer provided remarks to the editor, and did not raise any further technical concerns]

Reviewer #3

(Remarks to the Author)

The authors have properly addressed my concerns, now the acceptance is recommended.

Reviewer #4

(Remarks to the Author)

[Editor Note: This reviewer was asked to evaluate the rebuttal to Reviewer #2's comments.]

In this review comment, I mainly focus on the author's rebuttal and revision in response to flagged technical concerns. The authors have addressed majority comments from Reviewer 1 and Reviewer 2. However, for the concerns on inadequate novelty and unclear mechanism raised by Reviewer 3, I don't think the authors have effectively addressed these two concerns yet.

- In the response letter, the authors claim that "this dual control enables a synergistic enhancement in bulk cohesion and interfacial adhesion, which has not been achieved in prior studies", which is true. However, the authors didn't provide data to show how their approach synergistically enhances both bulk cohesion and interfacial adhesion. To make this mechanism clear, the authors are recommended to measure interfacial fracture toughness and interfacial fatigue threshold, which can help to decouple these two mechanisms.
- The authors show that the toughness and lap-shear strength of the proposed SST outperform existing thermal-resistant adhesives including commercially available epoxy-based adhesives. The summarized strength of epoxy-based adhesives in the manuscript is typically below 20 MPa as summarized in Figure 2 and Figure 5, which can be confusing. In fact, the existing commercially available epoxy-based adhesives work pretty well, which can achieve strength up to 70 MPa (Rudawska, Anna. "Mechanical properties of selected epoxy adhesive and adhesive joints of steel sheets." *Applied Mechanics* 2, no. 1 (2021): 108-126.). Thus, it is not convinced that the proposed SST truly outperforms epoxy-based adhesives.
- The authors attribute the thermal-resistant adhesion to the stability of ordered nanostructure at elevated temperature. Since the melting temperatures of most crystalline polymers are typically above 150 °C, it is hard to understand how stable adhesion performance across a temperature from 20 to 120 °C by SST outperforms existing polymer-based adhesives.
- The authors provided detailed information on thermal-resistant adhesion measurement following ASTM D1002, which is great. However, the explanation of the "thermal-resistant adhesion" test is unclear. The text states that samples were equilibrated in a heated chamber, but the actual shear tests were performed at room temperature. If the goal is to evaluate high-temperature adhesion, shear testing should be conducted under elevated temperatures, or the rationale for pre-heating followed by room-temperature testing must be clarified.
- The strategy using solution-sheared supramolecular oligomers can be a novel angle due to its advantage in structural control, processability, adhesive performance, and scalability. However, the authors didn't specifically compare this strategy to traditional polymer-based adhesive fabrications.

1) Precise Structural Control:

The manuscript highlights the ordered nanostructures formed via solution shearing of oligomers, but does not clarify whether similar effects are achievable in polymers, or why polymers are less favorable in this context.

2) Improved Processability:

The authors attribute better processability to the intrinsic features of oligomers (lower viscosity, reduced entanglement), but no data or comparison with polymers is provided. It also remains unclear whether these features are general or specific to their design.

3) Scalability:

The authors state that the process is scalable, yet present no experimental evidence or process discussion related to large-area coating, process uniformity, or industrial compatibility. All data appears limited to lab-scale, static shear coating.

Reviewer #5

(Remarks to the Author)

I co-reviewed this manuscript with one of the reviewers who provided the listed reports. This is part of the Nature Communications initiative to facilitate training in peer review and to provide appropriate recognition for Early Career

Researchers who co-review manuscripts.

Version 2:

Reviewer comments:

Reviewer #4

(Remarks to the Author)

The authors have adequately addressed my concerns. I would be delighted to recommend the publication of this work.

Reviewer #5

(Remarks to the Author)

Response to the reviewers' comments

Reviewer #1 (Remarks to the Author):

1. Please specify the chemical structure of the entire polymer at the early part of the manuscript. Also, include schemes that illustrate the polymer synthesis process. These should be presented in or before Figure 1 in the main text rather than in the supporting information.

Response:

Thank you for your insightful feedback. We have taken your suggestions into consideration and made several enhancements to improve the clarity and accessibility of our manuscript. Specifically, we have incorporated the complete chemical structure of the oligomers and a detailed scheme of the synthesis process directly into the main text. These additions are now prominently displayed in Figure 1a of the revised manuscript, ensuring that they are easily accessible and contribute effectively to the overall flow of the document. We believe these changes will provide our readers with a clearer understanding of the material from the outset.

a) Rational design of molecular building blocks

Figure R1. As a copy of Figure 1a shows that the chemical structures and the synthesis process of the resultant supramolecular oligomers.

2. The explanation of the multiscale adhesion mechanism looks good; however, are these demonstrated mechanisms hypotheses or proven scientific concepts? The detail discussion / explanation is required.

Response:

We sincerely appreciate the reviewer's acknowledgment of the multiscale adhesion mechanism in our work, which is a central focus in designing the supramolecular oligomers with simultaneous enhancement of strong and thermal-resistant interfacial adhesion and bulk cohesion. By utilizing controlled solution-shearing processes, we strategically regulate the orientations of hydrogen-bonded nanostructures that significantly improve the mechanical and adhesion properties of the supramolecular oligomers. This methodical structuring results in oriented nanocrystals and aligned nanofibrils which are absent in previous works.

As illustrated in Figure 1, our design strategy spans multiple scales—molecular, nanoscale, and microscale—to prepare solution-sheared supramolecular oligomer adhesives. At the molecular level, we incorporated hierarchical hydrogen bonding motifs (e.g., UPy, urea, and amide) alongside soft flexible blocks (e.g., priamine 1074 and adipic acid) to create multi-block oligomers. At the nanoscale, the orientation of nanocrystals and the alignment of nanofibrils were enabled by the solution shearing process, as referred to Figure 3 of the manuscript. Furthermore, at the microscale, the formation of aligned microfibrils within the bulk was validated by Figure S14. Finally, the oligomers demonstrated high mechanical robustness and strong and thermal-tolerant adhesion on diverse substrates.

We would like to clarify that the mechanisms discussed are not merely hypotheses but have been experimentally confirmed in our study.

(1) Enhanced Bulk Cohesion: We employed characterization techniques, including Differential Scanning Calorimetry (DSC), X-Ray Diffraction (XRD), Atomic Force Microscopy (AFM), and Small-Angle X-Ray Scattering (SAXS), to demonstrate the presence of **oriented nanostructures** (refer to Figure 3 of the revised manuscript). For example:

- DSC Analysis: an increase in melting temperatures and crystallinity at higher shear rates, indicating the formation of oriented structures.
- XRD and SAXS Results: an enhancement in crystallinity from 43% to 60% and significant growth in the sizes of nanocrystals, validating our claims about the impact of shearing on nanostructural properties.
- AFM phase images show the **aligned nanofibrils** of SST-2-10 (Figure 3c-d of the revised manuscript).
- SEM images indicate the formation of **microfibers** of SST-2-10 (Figure S14 of the SI).

(2) Improved Interfacial Adhesion: Our surface characterization included:

- ✧ Chemical Analysis: Temperature-dependent ATR-FTIR and Raman spectroscopy and mapping revealed **higher intensities of dissociated UPy motifs and ordered urea units** in sheared samples compared to unsheared ones, suggesting stronger interfacial bonding (Figure S18 and Figures 4f-g).
- ✧ Modulus Mapping and Surface Adhesion Force: Conducted using Peak Force Quantitative Nanomechanical Mapping, these tests indicated a **higher surface modulus and improved adhesion force** of SST-2-10 in contrast to the control sample, aligning with our hypothesis regarding the role of hydrogen-bonded nanocrystals (Figure 4h-i and Figure S19).

(3) Surface Interactions and Binding Motifs: The interactions at the adhesive interface were detailed using schematic illustrations (Figure 1b and Figure 4k) and supported by Raman analysis (Figure 4f-g), providing a clear depiction of the chemical components and their interactions, thus supporting our model of surface interactions.

(4) Multiscale Engineering Approach: We provided comprehensive schematics (Figure 1c) to illustrate how we integrate molecular, nanoscale, and microscale features to engineer the hierarchical structure of our adhesives. This multiscale approach is crucial for achieving the enhanced properties reported.

Figure R2. As a copy of Figure 1 showed that chemical structures and the synthetic route of supramolecular thermoplastic oligomers (a); Interfacial adhesion and bulk cohesion enabled by the solution-shearing of supramolecular thermoplastics (SST-n-m) that display ordered and enlarged nanocrystals and aligned nanofibrils in the bulk phase and anchored to the surface (b); Multiscale engineering used for the design and construction of supramolecular structural adhesives with high-strength and thermal-resistant adhesion (c).

Figure R3. As a copy of Figure 3 of the revised manuscript showed structural properties. (a) DSC curves of SST-2-m treated with the shear rate of 0, 5, 10, and 15 mm/s. (b) XRD profiles of SST-2-m on a glass plate and treated with the shear rate of 0, 5, 10, and 15 mm/s. The curves were recorded by the second heating scan from -60 to 180°C with a heating rate of 10°C/minutes. AFM phase images of the nanofibrils arrangement of SST-2-0 (c) and SST-2-10 (d) on the glass substrate. (e) The scattering intensity versus q vector parallel to nanofibrils of the shearing direction. (f) The scattering intensity versus q vector perpendicular to nanofibrils of the shearing direction. (g) The average distance between adjacent nanocrystals of sheared films parallel and perpendicular to the shearing direction at a shear rate of 0, 5, 10, and 15 mm/s. (h) The cross-sectioned schematic of the average distance between adjacent nanocrystals before and after shearing process.

Figure R4. As a copy of Figure S14 shows the formation of microfibrils of SST observed in SEM image.

Figure R5. As a copy of Figure 4 shows **Bulk and interfacial characterizations**. (a) Tensile stress-strain curves of SST-n-m recorded with a stretching rate of 10 mm/minutes. The tensile performance shown in this figure were performed in a direction parallel to the nanofibrils. Bar charts of the Young's modulus (b) and toughness (c) of SST-n-m. (d) Stress relaxation tests of SST-2-10 measured at different temperature. (e) The fitted curve of natural logarithm relaxation time versus temperature for SST-2-10. (f) Intensities of the disassociated UPy motifs and urea units tested by raman spectra and (g) mapping on the surface layers of SST-2-0 and SST-2-10. The surface adhesion force distributions and mapping of SST-2-0 (h) and SST-2-10 (i). (j) The improved and fracture-resistant cohesion is attributed to high energy ordered nanostructures that effectively pin cracks within the bulk. (k) High interfacial adhesion arises from anchored nanocrystals, which pin cracks both at the interface and within the bulk. These ordered nanostructures require higher energy for crack propagation compared to amorphous chains, thereby enabling fracture-resistant cohesion and adhesion. All the error bar represent the SD with at least three replicates.

Figure R6. As a copy of Figure S18 shows that stretching bands of C=O and N-H at the adhesive surface with the temperature changes.

Figure R7. As a copy of Figure S19 shows surface modulus measurements of SST-2-0 and SST-2-10, tested by AFM-based indentation.

(5) Experiments (e.g., stress relaxation, rheology, and viscosity) to quantify the **reversibility** and **dynamics** of the hydrogen-bonding interactions under varying temperature conditions.

Figure R8. As a copy of Figure 4d-e shows stress relaxation tests of SST-2-10 measured at different temperature.

Figure R9. As a copy of Figure 5d-e shows the temperature sweep rheological analysis of SST-2-10 at a constant frequency of 1.0 rad/s from 30 to 180 °C and variation of SST-2-10 with temperature between 120 to 180 °C measured during heating and cooling.

We hope that this additional information clarifies the experimental basis of our conclusions and demonstrates the rigorous nature of our research approach. We believe these data robustly support our hypotheses and contribute valuable insights into the field of adhesive materials.

3. On line 215, what do 020 and 110 in parentheses represent? Please specify these numbers.

Response:

We appreciate the reviewer's comment. To make it clear, we have marked the deconvoluted peaks in the XRD profiles. The peaks are centered at the 2 theta of 20.18° and 23.76°, assigned to the (020) and (110) planes of the crystalline methylene groups of adipic acid segments.

Figure R10. As a copy of Figure S12 shows that deconvoluted XRD peaks of SST-2-15.

4. What is the ratio among the three segments in the polymer?

Response:

We thank the reviewer for the insightful question. The ratios of the three segments in the oligomers were shown in Table S1 of the revised SI, as below. This ratio was carefully optimized to achieve a balance between interfacial adhesion and bulk cohesion, as demonstrated by the enhanced mechanical and adhesive properties.

Compound (mmol)	SST-1-0	SST-2-0	SST-3-0
UPy-NCO	2	2	2
Priamine 1074	2	3	4
Adipic acid	1	2	3

Table R1. As a copy of Table S1 shows that the ratios of the chemical compositions used in the oligomers.

5. If you determined this ratio, then why did you pick this particular ratio? Did you try other ratio? If not why not?

Response:

We sincerely thank the reviewer for raising this important point regarding the segment ratios in our oligomers design. The selection of the specific ratio for SST-2-0 was guided by the need to achieve an

optimal balance between interfacial adhesion, bulk cohesion, processability, and mechanical performance. The molar ratios of SST-2-10 are 2 eq. UPy-NCO, 3 eq. Priamine 1074, and 2 eq. adipic acid, which were chosen to optimize the interplay between hydrogen bonding, flexibility, and crystallinity of SST-2-0.

As demonstrated in Figure S6 and Figure 4a-c, SST-2-0 exhibits the greatest lap shear strength among the tested samples (SST-1-0, SST-2-0, and SST-3-0). This superior performance stems from an optimal balance between mechanical stiffness and toughness, which is critical for achieving strong adhesion while maintaining structural integrity.

Figure R11. As a copy of Figure S6 shows lap shear curves of SST samples, highlighting the superior performance of SST-2-10.

In this work, our primary focus was to investigate the effect of shear rate on the mechanical and adhesive properties during the sample preparation process. Therefore, we selected SST-2-0 as the primary testing sample due to its balanced properties. To provide a more comprehensive understanding, we have also done the adhesive measurements on the SST-1-m and SST-3-m, which were prepared at different shear rates. These results are presented in Figure R12 and Figure 2b, which shows the lap shear strength of SST-1-m and SST-3-m across varying shear rates. We can find that SST-2-10 exhibited superior adhesion strength among all the tested samples.

Figure R12. Lap shear strength of SST-1-m and SST-3-m that were prepared at different shear rates.

We acknowledge the reviewer's suggestion to explore segment ratios more systematically. Such studies further optimize the oligomers' performance and provide deeper insights into the structure-

property relationships. In future work, we plan to conduct a more systematic investigation of segment ratios to identify additional opportunities for performance enhancement.

6. Are the polymer segments randomly aligned, block, or exhibit any regularity? To avoid such fundamental questions, please discuss the polymer synthesis in detail.

Response:

We thank the reviewer for raising this important question regarding the alignment and regularity of the oligomer segments in our supramolecular oligomers. To address this concern, we have expanded the discussion on the oligomer synthesis and the resulting molecular structure in the revised manuscript.

The supramolecular oligomers were synthesized via polycondensation, as detailed in the Experimental Section. The molecular design incorporates three key components:

- (1) 2-Ureido-4-pyrimidinone (UPy), which provides strong multivalent hydrogen bonding for robust supramolecular assembly.
- (2) Adipic acid, which contributes to semicrystalline structures, enhancing mechanical properties of the oligomers.
- (3) Priamine 1074, a soft, flexible block that imparts flexibility and facilitates chain rearrangement.

The synthesis process involves the reaction of UPy-NCO with adipic acid and Priamine 1074 in a controlled manner, as shown in Figure 1a and described in Table S1. The reaction proceeds under nitrogen flow at 160°C, and the completion of the reaction is confirmed by the disappearance of the isocyanate peak at 2265 cm⁻¹ in the FTIR spectra (Figure S3). The resulting oligomers exhibit a semi-crystalline and microphase-segregated structure, as confirmed by XRD analyses (Figures S10-11).

Figure R13. As a copy of Figure S3 shows FTIR spectra of the SST adhesives.

Figure R14. As a copy of Figure S10 shows XRD profiles of the SST adhesives.

Figure R15. As a copy of Figure S11 shows crystallinity of the SST adhesives that was calculated by XRD profiles.

The polymer segments are not randomly aligned but rather form hierarchical nanostructures due to the controlled solution-shearing process. This process promotes the formation of enlarged, ordered nanocrystals and aligned nanofibrils within the bulk, as evidenced by AFM and SAXS analyses (Figures 3c-h). The alignment of these nanostructures is parallel to the direction of the applied shear, which enhances both bulk cohesion and interfacial adhesion.

In summary, the polymer segments exhibit a high degree of regularity and alignment due to the controlled synthesis and solution-shearing process.

7. The molecular weight information of the polymer should be specified in the main text. Any GPC (SEC) data?

Response:

Thank you for your valuable feedback regarding the molecular weight information of the oligomers. To address this comment, we have included the molecular weight data in the main text and provided additional details on the characterization methods.

The molecular weight of the resultant oligomers was determined using MALDI-TOF analysis. The mass weight data, provided in Figure S4 of the revised SI, confirms the presence of oligomers and provides detailed molecular weight distributions. For clarity, we have included Figure R16 (below) as a copy of Figure S4, which shows the molecular weight of SST-n-0 tested by MALDI-TOF using DCTB as a matrix.

Figure R16. As a copy of Figure S4 shows the molecular weight of SST-n-0 tested by MALDI-TOF using DCTB as a matrix.

We appreciate your suggestion regarding GPC data. While MALDI-TOF provides accurate molecular weight information for oligomers, we acknowledge that GPC could offer additional insights into the polydispersity index (PDI). To address this, we have conducted GPC measurements on the resultant oligomers, and the data is presented below (Figure R17). These results complement the MALDI-TOF analysis and provide a more comprehensive understanding of the molecular weight distribution and PDI of the oligomers.

Figure R17. GPC profiles of the resultant oligomers.

8. Is there any crystallinity change over time? If the crystallinity changes after a few days of storage, the polymer's overall properties could also change. Additionally, crystallinity can change at different temperature ranges. Test results or, at the very least, a well-thought-out explanation is required.

Response:

We thank the reviewer for raising this important question regarding crystallinity changes over time and temperature. To address this, we have conducted XRD analysis on SST-2-10 at room temperature over a period of 30 days, measuring crystallinity at specific time points. As indicated in Figure R18, the crystallinity of SST-2-10 witnesses no obvious changes during this period. This highlights the role of hydrogen bonding and hierarchical nanostructures in maintaining crystallinity. This result is also consistent with the high and stable adhesion performance of the oligomer, demonstrating its robustness over time.

Figure R18. The crystallinity of SST-2-0 measured at specific time points over 30 days.

We also acknowledge the reviewer's concern regarding the crystallinity changes at different temperature ranges. To clarify this, we performed temperature-resolved XRD on SST-2-10 using a temperature-controlled stage. The degree of crystallinity was calculated at various temperatures, and the results are presented in Figure S20. As expected, the crystallinity of SST-2-10 decreases at elevated temperature due to the dissociation of hydrogen bonds and the subsequent melting of nanocrystals. The results are presented below:

Figure R19. As a copy of Figure S20 shows the crystallinity of SST-2-10 as a function of temperature.

At 90°C, the crystallinity began to decrease but remained above 50%, resulting in a lap shear adhesion strength exceeding 25 MPa. At 130°C, with crystallinity above 20%, the adhesion strength remained above 15 MPa. The adhesion strength decreased to 5.3 MPa at 150°C, where crystallinity dropped to around 10%. This is attributed to the **melting of the nanocrystalline domains** derived from adipic acid segments.

These findings align with the observed changes in lap shear strength at elevated temperatures, as discussed at paragraph 1, page 14 in the revised manuscript,

“These findings are consistent with the observed changes in crystallinity at elevated temperatures (Figure S20). At 90°C, the crystallinity remained above 50%, resulting in a lap shear adhesion strength exceeding 25 MPa. At 130°C, with crystallinity above 20%, the adhesion strength remained above 15 MPa. At 150°C, where crystallinity dropped to around 10%, the adhesion strength decreased to 5.3 MPa.”

The temperature-dependent crystallinity data provide valuable insights into the material's thermal stability and performance under varying conditions.

9. The test method regarding solvent use and influence is somewhat confusing. Adhesion was tested after the complete drying of the solvent; what, then, is the purpose of using multiple solvents? To enhance clarity, the author should separate the solvent effect section. The current writing in the manuscript may cause confusion.

Response: Thank you for your comment regarding the solvent testing section. We want to clarify that the purpose of testing the adhesive's performance after exposure to various solvents was to evaluate its chemical resistance and durability under different environmental conditions, rather than to assess the effect of solvents during the adhesion process. The oligomer adhesives were fully dried before solvent exposure, ensuring that the solvent itself did not influence the adhesion process. Instead, the tests were designed to simulate real-world scenarios where the adhesive might come into contact with different solvents, such as water, acids, bases, or organic solvents, and to demonstrate its stability and robustness in such environments.

To enhance clarity and avoid confusion, we have separated the solvent resistance testing into a distinct subsection under the Results and Discussion section, titled "Chemical-resistant and Durable Adhesion" in the revised manuscript:

“To evaluate the chemical resistance and durability of the adhesive, the bonded samples were immersed in various solvents for 24 hours. The solvents included: polar solvents (water, saltwater, urea solution, acidic (pH = 1), and basic (pH = 14) solutions), non-polar solvents (hexane), and polar organic solvent (ethanol). The lap shear strength was measured to assess any changes in adhesion performance. The purpose of these tests was to determine the adhesive's stability and durability in environments where it might be exposed to different chemical conditions. This is particularly important for applications in harsh or variable environments, such as industrial, automotive, or biomedical settings. As shown in Figure S8, the adhesive maintained strong adhesion after exposure to all tested solvents, with only a minor reduction in lap shear strength observed in ethanol (retaining 86% of its revised strength). The stable adhesion in aqueous conditions is attributed to the robust assembly of hierarchical nanostructures and the shielding of water molecules by the hydrophobic alkyl chains. The resulting adhesives demonstrated excellent resistance to water, acids, bases, and non-polar solvents, highlighting its robustness and suitability for demanding applications.”

10. Please correct the typo in Figure S16's caption.

Response: Thank you for your careful reading. The typo in Figure 16's caption has been corrected.

11. In Figure S17, the term "Surface modulus on AFM" should be accompanied by brief background information.

Response:

We appreciate your suggestion and agree that additional context would be helpful for readers, and we have included a brief explanation in the revised manuscript to clarify this term.

The surface modulus refers to the mechanical properties of the oligomer adhesive's surface, such as stiffness or elasticity, measured using Atomic Force Microscopy (AFM). AFM determines the modulus by probing the surface with a sharp tip and analyzing the force-displacement response. This measurement provides critical insights into the mechanical behavior of the oligomer adhesives, which is essential for their application in areas such as coatings, membranes, and nanofabrication.

To improve the clarity and accessibility, we have included a brief explanation accompanying Figure S19 in the revised manuscript,

“We also investigated the surface stiffness or elasticity of the adhesive coatings using AFM-based indentation. The surface modulus was probed with a sharp tip and the force-displacement response was analyzed. This analysis provides valuable information about the mechanical properties of the adhesive surface, which are critical for understanding its performance in practical applications.”

12. Table S7 needs lines to specify the source.

Response: We acknowledge the reviewer's suggestion. The lines to specify the source has been added in Table S7.

13. What are the key reasons for strong adhesion at high temperatures?

Response: Thank you for your insightful question regarding the key reasons for strong adhesion at high temperatures. The strong adhesion of our solution-sheared supramolecular oligomers (SST) at elevated temperatures can be attributed to several key factors:

(1) **Dense Hydrogen-Bonding Networks:** The SST adhesives are designed with hierarchical hydrogen-bonding motifs, particularly utilizing 2-Ureido-4-pyrimidinone (UPy) units, which form strong multivalent hydrogen bonds. These bonds create a high density of physical crosslinks, contributing to the adhesive's mechanical strength and thermal stability.

(2) **Ordered Nanocrystals and Aligned Nanofibrils:** The solution-shearing process induces the formation of enlarged, ordered nanocrystals and aligned nanofibrils within the adhesive bulk and at the surface. The ordered nanocrystals, in particular, contribute to the thermal stability by maintaining structural integrity at elevated temperatures.

(3) **Enhanced Crystallinity:** The shearing process increases the crystallinity of the adhesive, as evidenced by DSC (Figure 3a) and XRD (Figure 3b) analyses. Higher crystallinity correlates with improved thermal resistance, as the crystalline regions provide additional stability and strength to the adhesive matrix, even under thermal stress.

(4) **High Activation Energy for Stress Relaxation:** The SST adhesives exhibit a high activation energy for stress relaxation, which is nearly 1.6 times higher than that of unsheared samples (Figure 4d). This high activation energy indicates restricted chain mobility and bond dissociation at elevated temperatures, further contributing to the adhesive's ability to maintain strong adhesion under thermal stress.

(5) **Thermal-resistant Nanostructures:** The presence of ordered nanocrystals and aligned nanofibrils enhances the adhesive's thermal resistance. These nanostructures prevent the adhesive from softening or degrading at high temperatures, allowing it to maintain strong adhesion even under extreme thermal conditions.

In summary, the strong adhesion of SST adhesives at high temperatures is a result of the combined effects of dense hydrogen-bonding networks, ordered nanocrystals, enhanced crystallinity, high activation energy for stress relaxation, and the overall nanostructural integrity imparted by the solution-shearing process. These factors work together to ensure that the adhesive remains robust and effective across a broad temperature range, making it suitable for demanding applications in high-temperature environments.

14. The term "Ultrahigh" should be removed from the title.

Response: Thank you for the reviewer's suggestion. The term "ultrahigh" has been removed from the title.

15. The author should provide an explanation of the "solution shared process." This concept is crucial to the paper but is not explained.

Response: We sincerely thank the reviewer for the valuable feedback regarding the solution-shearing process. We acknowledge that this concept is central to this work, and we have provided a more detailed explanation below to clarify its importance and methodology.

The solution-shearing process is a key technique used to control the molecular alignment and nanostructure formation within the supramolecular oligomers, ultimately enhancing both bulk cohesion and interfacial adhesion. Here's a step-by-step explanation of the process:

(1) **Preparation of the Supramolecular Oligomer Solution:** The supramolecular oligomers are dissolved in a suitable solvent (e.g., DMSO) at a specific concentration (e.g., 10 mg/mL). The solution is heated to ensure complete dissolution and homogeneity.

(2) **Deposition on a Substrate:** The prepared solution is drop-casted onto a preheated substrate (e.g., glass or metal) held at a controlled temperature (e.g., 160°C). The substrate is positioned at a slight tilt angle (e.g., 1°) to facilitate the shearing process.

(3) **Application of Shear Force:** A shearing plate is placed above the substrate with a fixed gap distance (e.g., $200 \pm 10 \mu\text{m}$). The shearing plate is then moved at a controlled speed (ranging from 0 to 15 mm/s) along a specific direction. This controlled movement applies a shear force to the solution, inducing molecular alignment and nanostructure formation.

(4) Formation of Ordered Nanostructures: The solution shearing process causes the oligomers to align along the direction of the applied force, leading to the formation of enlarged, ordered nanocrystals and aligned nanofibrils within the bulk of the adhesive. This alignment is crucial for enhancing the mechanical properties, such as tensile strength and toughness, as well as improving adhesion.

(5) Solvent Evaporation and Cooling: After shearing, the film is left on the heated substrate to allow for complete solvent evaporation. The film is then cooled to room temperature, resulting in a solid adhesive layer with the desired nanostructural features.

(6) Adhesive Properties: The solution-shearing process enhances the adhesive's performance by creating hierarchical nanostructures with dense hydrogen-bonding networks, ordered nanocrystals, and aligned nanofibrils. These features contribute to the adhesive's exceptional mechanical robustness, thermal stability, and strong adhesion to various substrates.

Importance of Solution-Shearing

The solution-shearing process is critical because it allows for precise control over the molecular and nanostructural organization of the adhesive. By aligning the oligomers and inducing the formation of ordered nanocrystals and nanofibrils, the process significantly enhances both **bulk cohesion (through improved mechanical properties)** and **interfacial adhesion (through increased binding sites and stress dissipation)**. This dual enhancement is essential for achieving the high-performance adhesive properties demonstrated in this work, particularly in high-temperature environments.

16. The concepts of H bonding in "nano aggregates" and "nanocrystals" must be thoroughly explained. These terms are used frequently in the manuscript without clear definitions.

Response: Thank you for your feedback regarding the concepts of hydrogen bonding in nanoaggregates and nanocrystals. We agree that these terms are central to the manuscript and should be clearly defined to ensure readers fully understand their significance.

The terms "nano aggregates" and "nanocrystals" refer to specific nanostructures formed within the supramolecular oligomers due to the hierarchical hydrogen-bonding interactions. These structures play a critical role in enhancing the adhesive's mechanical properties, thermal stability, and interfacial adhesion. To provide clarity, we have included a detailed definition and explanation of these terms, as outlined below:

(1) Nanoaggregates

Nanoaggregates are small clusters of molecules that form through non-covalent interactions, such as hydrogen bonding. In the context of this work, nanoaggregates refer to the self-assembled structures formed by the supramolecular oligomers due to the strong hydrogen-bonding interactions between functional groups (e.g., UPy units). This term has been widely reported in literatures (Prog. Polym. Sci. 2023, 142, 101689; Adv. Mater. 2020, 32, 191244; Angew. Chem. 2018, 57, 13838-13842; J. Am. Chem. Soc. 2014, 136, 6969-6977). The formation of hydrogen-bonded nanoaggregates enhances the adhesive's bulk cohesion by creating a network of physical crosslinks. These aggregates dissipate mechanical energy during deformation, improving the adhesive's toughness and resistance to crack propagation. Additionally, nanoaggregates at the interface provide abundant binding sites for strong adhesion to substrates.

(2) Nanocrystals

Nanocrystals are ordered, crystalline domains at the nanoscale that form within the adhesive matrix. In this study, adipic acid units contribute to the formation of nanocrystals within the supramolecular oligomers. The solution-shearing process promotes the growth and alignment of these nanocrystals. Nanocrystals contribute to the adhesive's mechanical robustness and thermal stability. The ordered crystalline structure provides strength and rigidity, while the alignment of nanocrystals along the shear direction enhances the adhesive's ability to withstand stress and deformation. At the interface,

nanocrystals act as anchoring points, improving adhesion by creating a strong mechanical interlock with the substrate.

In summary, the combination of nanoaggregates and nanocrystals creates a hierarchical structure that enhances both bulk cohesion and interfacial adhesion. The hydrogen bonds within nano aggregates provide reversible crosslinks that dissipate energy, while the nanocrystals offer structural integrity and thermal stability.

In the revised manuscript, we have added a dedicated subsection under the Results and Discussion section titled "**Hierarchical Nanostructures Designed by Hydrogen Bonding**" to provide a detailed explanation of these concepts, including their definitions, formation mechanisms, and roles in the adhesive's performance,

“In this work, the solution-sheared supramolecular oligomers featured the hierarchical nanostructures formed through hydrogen bonding. These nanostructures include nanoaggregates, nanocrystals, and nanofibrils, which play distinct yet complementary roles in enhancing bulk cohesion and interfacial adhesion: (1) Nanoaggregates are self-assembled clusters of oligomers driven by strong hydrogen-bonding interactions, particularly the quadruple hydrogen bonds formed by UPy units. The nanoaggregates create a network of physical crosslinks within the adhesive, improving energy dissipation and crack resistance. (2) Nanocrystals are ordered, crystalline domains formed by the adipic acid units within the oligomers. The solution-shearing process promotes the growth and alignment of nanocrystals, enhancing the adhesive's mechanical strength and thermal stability. (3) Nanofibrils are elongated, fiber-like structures that are oriented parallel to the direction of shear. These nanofibrils enhance the adhesive's mechanical robustness and toughness by dissipating stress during deformation and providing structural integrity. The synergistic effect of nanoaggregates, nanocrystals, and nanofibrils, stabilized by hierarchical hydrogen bonding, enables the SST adhesives to achieve unprecedented performance in terms of strength, toughness, and thermal resistance.”

17. How are "Hierarchical aggregates" formed? Please provide a clear definition.

Response: We thank the reviewer's comment regarding the formation of "hierarchical aggregates". We appreciate the opportunity to clarify that the term "hierarchical aggregates" was intended to refer to "hierarchical nanostructures", which include nanoaggregates, nanocrystals and nanofibrils. These nanostructures play a crucial role in enhancing thermal tolerance and mechanical properties of the adhesives. To enhance clarity, we have changed the term of "Hierarchical aggregates" to "Hierarchical nanostructures" in the revised manuscript.

The term "hierarchical nanostructures" refers to the multiscale organization of supramolecular oligomers into distinct nanoscale features, including nanoaggregates, nanocrystals, and nanofibrils. These structures are formed through a combination of molecular design and the solution-shearing process, as described below:

(1) Molecular Design: The supramolecular oligomers are designed with soft building blocks and strong hydrogen-bonding motifs (e.g., UPy units). The adipic acid motifs contribute to the formation of crystallization, while the soft blocks (e.g., Priamine 1074) provide flexibility and facilitate molecular rearrangement. The hydrogen-bonding motifs drive the self-assembly of oligomers into nanoaggregates, which are small clusters of molecules held together by non-covalent interactions.

(2) Solution-Shearing Process: During the solution-shearing process, the applied shear force aligns the oligomers and promotes the formation of ordered nanocrystals from the rigid blocks. These nanocrystals are crystalline domains at the nanoscale that provide structural integrity and thermal stability. The shear force also induces the alignment of oligomers into nanofibrils, which are elongated, fiber-like structures that enhance mechanical robustness and toughness.

(3) Hierarchical Organization: The combination of nanoaggregates, nanocrystals, and nanofibrils creates a hierarchical nanostructure within the adhesive. This multi-level organization allows for

efficient energy dissipation, crack resistance, and thermal tolerance. At the molecular level, hydrogen bonding drives the formation of nanoaggregates. At the nanoscale, the rigid blocks crystallize into nanocrystals, while the aligned oligomers form nanofibrils. This hierarchical arrangement ensures that the adhesive maintains its mechanical properties and adhesion strength even under extreme conditions.

In summary, the hierarchical nanostructures are formed through a synergistic combination of **molecular design** and the **solution-shearing process**, resulting in a material with exceptional thermal tolerance, mechanical robustness, and adhesion performance.

18. On line 111, adipic acid contains an aliphatic linkage between functional groups, meaning individual segments do not exhibit rigidity. Please delete the term "rigid." While adipic acid can form a crystallizable block, individual adipic segment does not confer rigidity.

Response: We appreciate the reviewer's insightful suggestion. We agree with your observation that adipic acid contains an aliphatic linkage between functional groups, and individual adipic segments do not exhibit rigidity. The adipic acid segments can contribute to nanocrystals due to the hydrogen bonding interactions, enabling mechanical robustness and thermal stability of the resulting oligomers. To improve the clarity and accuracy of this work, we have removed the term "rigid" used for definition of adipic acid in the revised manuscript. We sincerely thank the reviewer's attention to detail, which has helped enhance the precision of our manuscript.

19. Please define "ST polymer" and "SST" at least once in the manuscript before using these acronyms.

Response: Thank you for the reviewer's comment. We appreciate the opportunity to clarify the definitions of "ST polymer" and "SST" in the manuscript.

The term "ST" refers to supramolecular thermoplastic oligomers that are not prepared by the solution shearing process. These oligomers are synthesized via polycondensation and exhibit semi-crystalline and microphase-segregated structures.

While "SST" represents solution-sheared supramolecular thermoplastics, which are prepared using this method. The acronym "SST-n-m" is used to denote these materials, where n represents the feed quantity of adipic acid (ranging from 1 to 3), and m represents the applied shear rate (ranging from 0 to 15 mm/s). Specifically, "SST-n-0" refers to oligomers that are not prepared by solution shearing.

To enhance clarity, we have added these definitions in the revised manuscript at paragraph 1, page 6, with the changes marked in red. We hope this clarification addresses the reviewer's concern and improves the readability of the manuscript.

20. Can this work be applied in real-world scenarios? The presented control parameters, such as solvent evaporation, temperature control, and shearing control, appear too delicate for practical applications.

Response:

We sincerely appreciate the reviewer's insightful comment on the practical application of the control parameters presented in our study, such as solvent evaporation, temperature control, and shearing control. Your observation allows us to further elaborate on the feasibility and relevance of these parameters in real-world scenarios.

While these parameters may indeed seem delicate, they are integral to numerous industrial processes, especially in the synthesis of advanced materials and chemical manufacturing. For example, the production of high-performance polymers and nanoparticles frequently necessitates precise control over temperature and solvent evaporation, which is typically managed using advanced equipment such as precision reactors or microfluidic systems. Similarly, shearing control is a fundamental aspect of processes like inkjet printing and the formulation of complex fluids such as paints and coatings.

In terms of practical applicability, the deployment of our methods in industrial settings hinges on the specific industry's ability to adopt technologies that can sustain such precise conditions. Scaling these processes to commercial levels may involve significant engineering advancements to ensure robustness and cost-effectiveness. Automated systems equipped with real-time monitoring and feedback controls could potentially streamline these processes, thereby minimizing variability and enhancing production efficiency.

Furthermore, the underlying principles elucidated in our research—particularly the roles of hierarchical nanostructures and hydrogen bonding in improving adhesion and thermal stability—offer a solid foundation for the development of scalable and practical applications. We acknowledge, as the reviewer highlighted, that transitioning from laboratory precision to industrial-scale applicability remains a formidable challenge, and it is one that warrants dedicated investigation and innovation.

By continuing to refine these processes and technologies, we aim to bridge the gap between laboratory research and practical, real-world applications, ultimately facilitating the broader adoption and implementation of these advanced materials.

21. The influence of the applied shear rate on adhesion properties is well-established textbook knowledge and has been reported extensively in adhesion and polymer science for decades. Reporting this effect may be redundant.

Response:

We sincerely thank the reviewer for their valuable feedback regarding the influence of shear rate on adhesion properties. We acknowledge that the relationship between shear rate and adhesion is well-established in adhesion and polymer science. However, our work advances beyond this fundamental understanding by demonstrating how **solution-shearing** can **simultaneously control** both the **adhesion chemistry at the interface** and the **nano/microstructure of the adhesive layer**. This dual control represents a significant advancement, as it enables the development of **thermal-resistant hydrogen-bonded adhesives** with unprecedented performance in **bulk cohesion** and **interfacial adhesion**.

By leveraging solution-shearing, we have established a new benchmark for such adhesives, offering a **scalable** and **versatile approach** to optimizing mechanical, thermal, and adhesive properties. Our findings not only reinforce the known importance of solution-shearing process but also provide a **novel framework** for designing next-generation adhesive materials with tailored properties for demanding applications.

To improve the clarity and focus of this work, we have revised the manuscript to more clearly emphasize these unique contributions. Specifically, we have framed the discussion of shear rate within the context of its innovative application, as outlined in the revised manuscript:

“While the relationship between shear rate and adhesion properties is indeed fundamental, our work advances beyond this by demonstrating how solution-shearing can simultaneously control both the adhesion chemistry at the interface and the nano/microstructure of the adhesive layer. This dual control is a significant advancement, as it enables the development of thermal-resistant hydrogen-bonded adhesives with unprecedented performance in bulk cohesion and interfacial adhesion. By leveraging solution-shearing, we have established a new benchmark for such adhesives, offering a scalable and versatile approach to optimizing mechanical, thermal, and adhesive properties. Our findings not only reinforce the known importance of solution-shearing process but also provide a novel framework for designing next-generation adhesive materials with tailored properties for demanding applications.”

22. This manuscript contains extensive benchmarking tests against commercial adhesives and previously reported adhesives. However, the main scientific or engineering discovery from this study is unclear. The manuscript appears more suitable for an engineering and/or application-focused

journal.

Response:

We sincerely thank the reviewer for their thoughtful feedback regarding the extensive benchmarking tests included in our manuscript. We appreciate the opportunity to clarify the **scientific and engineering advancements** that underpin this work.

While the benchmarking against commercial and previously reported adhesives highlights the **practical performance** of our material, the key discovery of this work lies in the development of a **solution-shearing** strategy that enables the **growth and orientation of nanocrystals** and the **alignment of nanofibrils** in hydrogen-bonded supramolecular oligomers. Importantly, this strategy simultaneously controls both the **adhesion chemistry at the interface** and the **nano/microstructure of the adhesive layer**. This dual control enables a **synergistic enhancement** in both **bulk cohesion** and **interfacial adhesion**, which represents a significant departure from conventional approaches that often optimize these properties separately.

Furthermore, our work establishes a new framework for designing **thermal-resistant hydrogen-bonded adhesives** with tailored properties, offering insights into the interplay between **processing conditions, material structure, and performance**. These findings are not only relevant to adhesive applications but also provide broader implications for the design of advanced functional materials.

We acknowledge that our work has strong **application-focused elements**, but we believe the **fundamental scientific principles** and **engineering strategies** presented here are equally impactful and broadly relevant to the materials science community. To better highlight these contributions, we have revised the manuscript to more clearly articulate the scientific and engineering contributions, ensuring they resonate with both the engineering and scientific sectors of the materials science field.

23. While the manuscript presents high-temperature-resistant adhesives, numerous reports already exist on such materials. Moreover, the mechanism based on hydrogen bonding from known functionalities (e.g., Upy) is well-documented. The author uses vague terms but fancy terms such as "solution shearing process" and "Hierarchical aggregates," which are not clearly defined in the manuscript. These terms might mislead editors and readers by describing well-known textbook knowledge with new and unclear terminology. I have serious concerns regarding the novelty of this manuscript.

Response:

We sincerely thank the reviewer for their thorough evaluation and constructive feedback. We appreciate the opportunity to address the concerns regarding the novelty of our work and the use of term such as **solution-shearing process** and **hierarchical nanostructures**. While we acknowledge that high-temperature-resistant adhesives and hydrogen-bonding mechanisms (e.g., based on UPy functionalities) have been extensively studied, our work introduces several **unique advancements** that distinguish it from prior reports:

(1) **Novelty of the Solution-Shearing Process.** While hydrogen bonding and its role in adhesion are well-documented, our solution-shearing process represents a novel approach to simultaneously control both the **interfacial adhesion chemistry** and the **nano/microstructure** of the adhesive layer. Specifically, the solution-shearing process aligns oligomers along the direction of the applied force, leading to the formation of **enlarged, ordered nanocrystals** and **aligned nanofibrils** within the bulk adhesive. This dual control enables a synergistic enhancement in bulk cohesion and interfacial adhesion, which has not been achieved in prior studies.

(2) **Hierarchical nanostructures.** The term "hierarchical nanostructures" refers to the multi-scale structural organization of the adhesives, including **nanoaggregates, nanocrystals, and nanofibrils**, driven by the hierarchical hydrogen-bonding interactions. At the molecular level, hydrogen bonding drives the formation of nanoaggregates. At the nanoscale, the adipic acid blocks crystallize into nanocrystals, while the aligned oligomers form nanofibrils. This multi-level organization enables

efficient energy dissipation, crack resistance, and thermal tolerance, ensuring that the adhesive maintains its mechanical properties and adhesion strength even under extreme conditions.

(3) **Advancements Beyond Literature Reports.** While we build on established principles of hydrogen bonding, our work extends these concepts by demonstrating how processing conditions (e.g., **solution-shearing**) can be leveraged to tailor material properties in a way that has not been previously reported. This approach offers a new framework for designing adhesives with enhanced thermal and mechanical performance. As highlighted in Table S7 of the revised SI, our work demonstrates a significant enhancement in the performance of hydrogen-bonded thermal-resistant adhesives.

Materials	Temperature	Shear stress	Substrate type	Source
SST-2-10	50°C	32 MPa	Stainless steel	This work
	70°C	33.8 MPa	Stainless steel	
	90°C	29.8 MPa	Stainless steel	
	95°C	28.7 MPa	Stainless steel	
	110°C	24.7 MPa	Stainless steel	
	120°C	21.1 MPa	Stainless steel	
	130°C	16.1 MPa	Stainless steel	
	140°C	10.1 MPa	Stainless steel	
	150°C	4.9 MPa	Stainless steel	
	110°C	24.8 MPa	Aluminum	
	90°C	30.4 MPa	Glass	
110°C	25.5 MPa	Glass		
Epoxy (Loctite Quick Set)	95°C	2 MPa	Stainless steel	Commercial
Poly(UPy-HMDI-HEMA-co-hexyl-MA)	60°C	2.5 MPa	Stainless steel	ACS Appl. Mater. Interfaces 2015, 7, 13395–13404
Poly(UPy-HMDI-HEMA-co-butyl-MA)	60°C	4.1 MPa	Stainless steel	
20 wt % SiNPs (wet)	95°C	4.1 MPa	Stainless steel	Sci. Adv. 2021, 7, eabk2451
20 wt % SiNPs (dry)	95°C	11.4 MPa	Stainless steel	
IC gel	60°C	4.5 MPa	Glass	Angew. Chem. 2021,60,8948–8959
	90°C	0.4 MPa	Glass	
	120°C	0.1 MPa	Glass	
P(nBuA-co-Ba-co-HW)	60°C	12 MPa	Glass	Angew. Chem. 2022, e202203876
	90°C	6.5 MPa	Glass	
PUD20	45°C	4.2 MPa	Aluminum	Mater. Horiz., 2023, 10, 4183-4191
	60°C	1 MPa	Aluminum	
^{DC} MIN	70°C	7.1 MPa	Brass	Angew. Chem. 2024, 63, e202409705
Ionogels	85°C	0.5 MPa	Glass	J. Am. Chem. Soc. 2024, 146, 13903–13913

Table R2. As a copy of Table S7 shows the benchmarking with the reported hydrogen-bonded adhesives at high temperature, further underscoring the advancements achieved in this work.

To address the reviewer’s concerns, we have revised the manuscript to more clearly define and elaborate on these key concepts, ensuring that the **novelty** and **scientific contributions** of our work are presented in a way that is both accurate and accessible to readers. We sincerely appreciate the reviewer’s valuable feedback, which has helped us improve the clarity and impact of our manuscript.

Reviewer #2 (Remarks to the Author):

Designing strong and thermally resistant adhesives has become a key research focus for load-bearing

applications. The authors present a strategy involving solution-sheared supramolecular oligomers, which exhibit excellent adhesive strength and toughness across a broad temperature range. This adhesive demonstrates impressive lap shear strength, surpassing that of commercial structural and thermoset adhesives. However, several important issues need to be properly addressed before the manuscript can be considered for publication.

Response:

We sincerely appreciate the reviewer's thoughtful evaluation and positive feedback to our work. We appreciate your recognition of the potential of our solution-sheared supramolecular oligomers as strong and thermally resistant adhesives for load-bearing applications. Below, we have provided a detailed response to the reviewer's concerns.

1. Why did the authors choose the solution-sheared supramolecular oligomer strategy? Why opt for an oligomer instead of a polymer?

Response:

Thank you for your insightful question. We appreciate the opportunity to clarify our choice of the solution-sheared supramolecular oligomer strategy and the rationale behind using oligomers instead of polymers.

The solution-sheared supramolecular oligomer strategy was selected for its ability to deliver precise structural control, improved processability, and enhanced performance.

(1) **Precise Structural Control.** The solution-shearing process enables precise alignment and organization of oligomers, leading to the formation of ordered nanocrystals and aligned nanofibrils. This hierarchical nanostructure enhances both bulk cohesion and interfacial adhesion, which is critical for achieving high-performance adhesives.

(2) **Improved Processability.** Oligomers, being smaller and less entangled than polymers, offer greater processability under shear conditions. Their lower viscosity and reduced entanglement facilitate uniform alignment, making the solution-shearing process highly scalable and suitable for industrial applications.

(3) **Superior Performance.** The combination of solution-shearing and supramolecular interactions (e.g., hydrogen bonding) allows for the creation of adhesives with exceptional strength, toughness, and thermal stability. The higher density of functional groups in oligomers (e.g., UPy units) enhances supramolecular interactions, resulting in improved energy dissipation and resistance to deformation under stress.

(4) **Scalability and Versatility.** The solution-shearing process is compatible with a wide range of substrates. Oligomers, with their lower viscosity and better solubility, enhance their processability, making this approach practical for various applications.

In summary, our choice of oligomers and the solution-shearing strategy is driven by their ability to provide precise structural control, enhanced processability, and superior performance, making them ideal for developing high-performance, thermally resistant adhesives.

2. The quality of the ^1H NMR spectrum data for ST-2 is low. Some proton peaks remain unassigned.

Response:

We sincerely thank the reviewer for the careful evaluation and for pointing out the issue with the quality of the ^1H NMR spectrum for SST-2-0. To address this, we have re-examined the spectrum and enhanced its quality. Due to the abundance of alkyl units from Priamine 1074 and adipic acid, some proton peaks overlap, making it challenging to assign all peaks definitively. However, we have ensured that the key peaks corresponding to the functional groups (e.g., UPy units) are clearly resolved and assigned. Additionally, the disappearance of the proton peak corresponding to the COOH group of adipic acid at 12 ppm in the final oligomer confirms the completion of the

polycondensation reaction. This observation provides strong evidence for the successful synthesis of SST-2-0. The updated spectrum is presented below:

Figure R20. As a copy of Figure S2 shows the ^1H NMR spectrum of ST-2 recorded in $\text{CHCl}_3\text{-d}$.

3. The authors describe SST as a polymer film. However, this contradicts the MALDI-TOF mass spectrometry results, which suggest it is an oligomer with low molecular weight.

Response:

Thank you for your careful observation regarding the description of SST and the MALDI-TOF mass spectrometry results. We appreciate the opportunity to clarify that the term “polymer film” was intended to describe the macroscopic form of the SST material after processing, rather than to imply it is a high-molecular-weight polymer. We recognize that this wording may have caused confusion, as the MALDI-TOF mass spectrometry results clearly indicate that SST is an oligomer with a low molecular weight.

To avoid any misunderstandings, we have revised the manuscript to consistently refer to SST as an “oligomer film”, which more accurately reflects its molecular nature.

4. Was the failure mode during adhesion tests cohesive failure, adhesive interfacial failure, or a combination of both? Proper experimental evidence and discussion should be provided. Additionally, SST-2-10 exhibits outstanding lap shear strength, surpassing that of many commercial adhesives. The authors should conduct a cost analysis to assess its feasibility as a next-generation adhesive material.

Response: We thank the reviewer’s insightful comments and suggestions. To better understand the failure mode of adhesives, we have done additional adhesion experiments on copper surfaces. The copper substrates were bonded with a sheared film thickness of ca. 200 μm and a bonded area of 9 mm \times 15 mm. As indicated in Figure R21, a combination of cohesive and adhesive failure was observed in SST-2-n oligomers, indicating a simultaneous enhancement of interfacial adhesion and bulk cohesion of the sheared oligomer.

Oligomer	SST-2-0	SST-2-5	SST-2-10
Failure mode	Mixed failure	Mixed failure	Mixed failure
Failed lap joints			

Figure R21. The failure mode and failed lap joints of SST-2-n sandwiched by copper substrates.

We appreciate your suggestion to conduct a cost analysis to assess the feasibility of SST-2-10 as a next-generation adhesive material. While our current study focuses on the scientific and performance aspects, we recognize the importance of economic considerations for practical applications. Below, we have done a cost analysis comparing the developed adhesive (SST-2-10) with three widely used commercial adhesives: polyurethane, epoxy, and cyanoacrylate.

(1) Raw Material Costs

- SST-2-10:
 - The synthesis of SST-2-10 involves supramolecular oligomers with functional groups such as Upy (ureidopyrimidinone) and other hydrogen-bonding motifs. These raw materials are specialty chemicals, which are relatively expensive due to their high purity and functionalization requirements.
 - Estimated raw material cost: 50 – 100 per kilogram (depending on scale and supplier).
- Polyurethane Adhesives:
 - Polyurethane adhesives are derived from widely available petrochemical precursors such as isocyanates and polyols. These raw materials are produced at large scales, resulting in lower costs.
 - Estimated raw material cost: 5 – 20 per kilogram (depending on grade and supplier).
- Epoxy Adhesives:
 - Epoxy adhesives are based on epoxy resins and hardeners (e.g., amines or anhydrides). The raw materials are moderately priced and widely available.
 - Estimated raw material cost: 10 – 30 per kilogram (depending on formulation and supplier).
- Cyanoacrylate Adhesives:
 - Cyanoacrylate adhesives are derived from cyanoacrylate monomers, which are relatively inexpensive but require precise manufacturing processes.
 - Estimated raw material cost: 20 – 50 per kilogram (depending on purity and supplier).

(2) Processing Costs

- SST-2-10:
 - The solution-shearing process requires precise control of temperature, shear rate, and solvent evaporation. While scalable, it may involve higher energy and equipment costs compared to conventional adhesive manufacturing.
 - Estimated processing cost: 10 – 30 per kilogram.
- Polyurethane Adhesives:
 - Polyurethane adhesives are typically produced using bulk polymerization processes, which are energy-efficient and cost-effective at large scales.
 - Estimated processing cost: 2 – 10 per kilogram.

- Epoxy Adhesives:
 - Epoxy adhesives are produced using simple mixing and curing processes, which are relatively low-cost.
 - Estimated processing cost: 5 – 15 per kilogram.
- Cyanoacrylate Adhesives:
 - Cyanoacrylate adhesives require precise polymerization and stabilization processes, which can be more costly.
 - Estimated processing cost: 10 – 25 per kilogram.

3. Total Estimated Cost and Performance Comparison

Parameter	SST-2-10	Polyurethane	Epoxy	Cyanoacrylate
Raw Material Cost	50–100 per kg	5–20 per kg	10–30 per kg	20–50 per kg
Processing Cost	10–30 per kg	2–10 per kg	5–15 per kg	10–25 per kg
Total Cost	60–130 per kg	7–30 per kg	15–45 per kg	30–75 per kg
Key Advantages	Exceptional strength, thermal resistance, toughness	Low cost, flexibility, durability	High strength, chemical resistance	Fast curing, high bond strength
Limitations	Higher cost	Limited thermal stability	Brittle at low temperatures	Poor thermal and impact resistance

4. Performance vs. Cost Trade-Off

- SST-2-10:
 - While SST-2-10 is more expensive than commercial adhesives, its **outstanding lap shear strength, thermal resistance, and toughness** justify the higher cost for specialized applications where performance is critical (e.g., aerospace, automotive, or high-temperature industrial applications).
 - For bulk or cost-sensitive applications, further optimization of the synthesis and processing of SST-2-10 could reduce costs. For example, scaling up production, sourcing cheaper raw materials, or simplifying the solution-shearing process could make SST-2-10 more competitive.
- Commercial Adhesives:
 - Polyurethane, epoxy, and cyanoacrylate adhesives are cost-effective and widely used for general-purpose applications. However, they may lack the necessary thermal stability and mechanical performance for demanding environments.

5. Feasibility as a Next-Generation Adhesive

- SST-2-10 is highly feasible for high-performance applications where superior mechanical and thermal properties are required, even at a higher cost. However, for broader commercial adoption, cost reduction strategies will be essential. Future work will focus on optimizing the production process and exploring alternative raw materials to improve cost-effectiveness without compromising performance.

This cost analysis provides a balanced perspective on the feasibility of SST-2-10 as a next-generation adhesive material.

5. The adhesion mechanisms of the developed adhesive on various substrates should be more thoroughly characterized and discussed.

Response:

Thank you for your insightful comment regarding the adhesion mechanisms of our developed adhesive on various substrates. To address this, we have conducted additional characterization and discussion from both molecular and nanostructural perspectives to provide a more thorough understanding of the adhesion mechanisms.

(1) Molecular interactions at the surface.

The adhesion mechanism is driven by hydrogen bonding interactions between the adhesive and diverse substrates, as shown in Figure 4k of the revised manuscript. For example, the C=O groups from UPy and urea motifs of the adhesives interact with Si-OH units on glass surface and OH groups on epoxy surface; the NH groups from UPy and urea motifs bond with -F groups of Teflon surface and metal oxide (MO_x) groups on metal substrates.

To confirm the hydrogen bonding behaviour, ATR-FTIR spectral analysis was conducted on the adhesives surface. As shown in Figure S18 of the SI, upon heating from 30 to 130°C, the C=O stretching band gradually shifts from 1654 to 1670 cm⁻¹, while the N-H band shifts increases from 3328 to 3350 cm⁻¹. These shifts reveal a weakening of hydrogen bonding interactions at elevated temperature, confirming the role of hydrogen bonding in adhesion.

Figure R22. The schematic figure of hydrogen bonding adhesion mechanism in this work.

Figure R23. As a copy of Figure S18 shows stretching bands of C=O and N-H at the adhesive surface with the temperature.

(2) Nanostructures at the surface.

We also investigated the role of nanostructures in adhesion force and modulus using AFM-based indentation (Figure S19 and Figure 4h-i of the revised manuscript). The results show that the adhesive surface exhibits a **high modulus**, which is attributed to the accumulation of **nanocrystals** at the surface. These nanocrystals enhance the mechanical interlocking and cohesion of the adhesive, contributing to the overall adhesion strength.

Figure R24. As a copy of Figure S19 of the SI shows the surface stiffness and nanocrystal accumulation tested by AFM-based indentation.

Therefore, the combination of **abundant hydrogen bonding interactions** at the interface and the **nanocrystal-rich surface structure** synergistically improves the adhesion strength of the adhesive on various substrates.

6. In Figures 2i and 5e, literature sources should be cited for performance comparisons with previous works. Additionally, detailed testing conditions for the reported thermal-resistant adhesives should be provided.

Response:

Thank you for your careful reading. To address the reviewer’s comment, we have added the literature sources for performance comparisons, as indicated in Figure 2i and 5e of the revised manuscript.

Figure R25. As a copy of Figure 2i shows the benchmarking with the reported adhesives both in work of debonding and adhesive strength.

Figure R26. As a copy of Figure 5f shows the comparisons with the reported thermal-resistant adhesives.

We also appreciate the reviewer’s comment for providing detailed testing conditions for thermal-resistant adhesion in the method sections of the revised manuscript, as shown below:

“Thermal-resistant adhesion measurement. The lap joints were equilibrated in a controlled temperature chamber for 10 minutes to ensure thermal stabilization at the target temperature. Single-lap shear strength tests were performed using an INSTRON-5566 universal testing machine, following the ASTM D1002 standard. Tests were conducted at room temperature with a strain rate of 10 mm/min. The substrates, with a thickness of 2 mm, were bonded using a sheared adhesive film with a thickness of approximately 200 μm and a bonded area of 9 mm \times 9 mm. The lap shear stress was measured in a direction parallel to the oriented nanofibrils. Lap shear strength was calculated as the maximum force divided by the overlap area of the adhesive film. Reported shear strength values represent the average of three samples, with standard deviations provided for statistical reliability.”

7. An AFM topography image of SST-2-10 should be included to further support its aligned nanofibril structure.

Response: We agree with the reviewer’s valuable suggestion. To strengthen our manuscript, we have included an AFM topography image of SST-2-10, as below shown. This image provides clear visual evidence of the aligned nanofibril structure, supporting the hierarchical organization of the adhesive.

Figure R27. The AFM topological image of SST-2-10.

8. The authors should verify the frequency values in the AFM adhesion force measurements and consider providing fitted Gaussian distribution curves.

Response:

We thank the reviewer’s comment regarding the AFM adhesion force measurements. The adhesion force measurements were conducted at a tip oscillation frequency of 2 kHz, as stated in the revised manuscript. We have verified the frequency values and confirmed their accuracy in the data acquisition process.

We have included the Gaussian distribution curves for the adhesion force data, as indicated in Figure 4h-i of the revised manuscript. These figures show the adhesion force distributions for both SST-2-0 and SST-2-10.

Figure R28. As a copy of Figure 4h-i shows that adhesion force distributions of SST-2-0 and SST-2-10.

9. The response time of SST-2-10's mechanical properties to temperature variations should be discussed in Figure 5.

Response:

We sincerely thank the reviewer for their valuable suggestion. To address the comment regarding the response time of SST-2-10's mechanical properties to temperature variations, we have expanded the discussion in Figure 5 of the revised manuscript, focusing on the material's dynamic response to thermal changes.

As shown in Figure 4d of the revised manuscript, SST-2-10 exhibited distinct stress relaxation behaviors at different temperatures. The **relaxation time (τ)**, defined as the time required for the stress to decay to 1/e (37%) of its initial value, was found to decrease significantly with increasing temperature. At low temperatures, the topological network of SST-2-10 is effectively "frozen," resulting in slower stress relaxation and longer relaxation times. In contrast, at elevated temperatures, the relaxation process is significantly accelerated due to the increased mobility of the supramolecular network. This fast response time is attributed to the **reversible nature of the hydrogen bonding interactions** within the supramolecular network, which allows the material to rapidly reorganize its structure in response to thermal stimuli. Furthermore, the **hierarchical nanostructures** of SST-2-10, comprising nanocrystals and aligned nanofibrils, facilitates efficient energy dissipation and structural reorganization, contributing to its enhanced thermal responsiveness.

The response time of SST-2-10 to temperature variations encompasses not only stress relaxation but also the material's ability to reorganize its structure and adapt its mechanical properties.

The hierarchical nanostructure of SST-2-10, comprising nanocrystals and aligned nanofibrils, plays a critical role in enabling this rapid response. The ordered nanocrystals and aligned nanofibrils, formed through the solution-shearing process, facilitate efficient energy dissipation and structural reorganization, allowing the material to quickly adapt to temperature changes while maintaining its adhesive strength and toughness.

In the revised manuscript, we have included a detailed analysis of the response time in Figure 5, where we discuss the **temperature-dependent mechanical behavior** of SST-2-10. The modulus analysis also revealed a gradual decrease from 20°C to 130°C, followed by a sharp decline between 130°C and 180°C, coinciding with the phase transition of **hierarchical nanostructures** within SST-2-10. Specifically, we highlight how the material maintains its adhesive strength across a broad temperature range (20°C to 120°C), with a peak lap shear strength of 33.8 MPa at 70°C. This optimal performance at 70°C is attributed to the rubbery state of the UPy-functionalized crosslinks, which provide network flexibility and toughness at this temperature. Additionally, we discuss the material's ability to retain over 20 MPa of lap shear strength even after prolonged exposure to 120°C for 30 days, demonstrating its exceptional thermal stability and durability.

Figure R29. As a copy of Figure 5 of the revised manuscript shows **Thermal-resistant adhesion**. (a) The thermal-resistant adhesion mechanism is attributed to the stability of nanocrystalline domains at elevated temperature, even up to 130°C. (b) The lap shear strength of SST-2-10 tested at the ambient temperature ranging from 20 to 160°C. (c) The lap shear strength of SST-2-10 bonded between copper substrates after 10 cycles conducted at 50°C and 150°C, respectively. (d) The temperature sweep rheological analysis of SST-2-10 at a constant frequency of 1.0 rad/s from 30 to 180 °C. (e) Variation of SST-2-10 with temperature between 120 to 180°C measured during heating and cooling. (f) Comparison of the lap shear strength of SST-2-10 and the reported adhesives over a wide range temperature.

The revised discussion in Figure 5 also emphasizes the role of the **hierarchical nanostructures in enabling the material's rapid response to temperature variations**. The ordered nanocrystals and aligned nanofibrils, formed through the solution-shearing process, enhance the material's mechanical robustness and thermal tolerance, allowing it to maintain high performance under varying thermal conditions. This detailed analysis underscores the importance of the hierarchical nanostructures in achieving both rapid thermal responsiveness and long-term thermal stability.

We believe these additions provide a more comprehensive understanding of the response time of SST-2-10's mechanical properties to temperature variations and further highlight the material's potential for applications in demanding thermal environments.

We have conducted additional experiments and added detailed discussion in the revised manuscript, ensuring that the novelty and scientific contributions of our work are presented in a way that is both accurate and accessible to readers. Thank you again for your valuable feedback, which help us improve the clarity and impact of our manuscript.

Reviewer #3 (Remarks to the Author):

This work describes a class of solution-sheared supramolecular oligomers with strong adhesive strength and toughness across a broad temperature range. However, this work is a moderate extension of previous reported adhesives composed of UPy and/or urea units, there is not much novelty within. This mechanism explanation is not clear enough, the obtained data is insufficient and superficial. There is also no cutting-edge findings or striking progress towards the field of H-bonded adhesives. Therefore, this results appear to be more suitable for publication in a journal like Polymer Journal, does not appear to be of enough urgency for high-ranking Nature Communications. Moreover, there is a significant issue need to be clarified, in line 388-399, the authors mention that "Solution-shearing of supramolecular thermoplastics (SST). The SST solution was prepared in DMSO at a concentration of 10 mg/mL and then was heated in a glass dish at 160°C for 30 min. left on the heating plate at 60°C for 12 hours until the residual solvent was completely removed." We know that the boiling point of DMSO is as high as 189 oC, this reviewer is wondering how does the residual DMSO could be completely removed by heating 160 or 60 oC, the evidence regarding the complete removing of DMSO must be provided.

Response:

Thank you for your thorough evaluation and constructive feedback. We appreciate your acknowledgment of the strong adhesive strength and toughness of our solution-sheared supramolecular oligomers across a broad temperature range. Below, we address the reviewer's concerns regarding novelty, mechanistic clarity, data depth, and solvent removal, and outline the revisions made in the revised manuscript.

Novelty and Advancements:

While supramolecular adhesives based on UPy and urea units have been previously reported, our work introduces several unique advancements that distinguish it from prior reports:

(1) **Novelty of the Solution-Shearing Process.** While hydrogen bonding and its role in adhesion are well-documented, our solution-shearing process represents a novel approach to simultaneously control both the interfacial adhesion chemistry and the nano/microstructure of the adhesive layer, as indicated in Figure 1 of the revised manuscript. Specifically, the solution-shearing process aligns oligomers along the direction of the applied force, leading to the formation of **enlarged, ordered nanocrystals** and **aligned nanofibrils** within the bulk adhesive. This dual control enables a **synergistic enhancement in bulk cohesion and interfacial adhesion**, which has not been achieved in prior studies.

(2) **Hierarchical nanostructures.** The term “hierarchical nanostructures” refers to the multi-scale structural organization of the adhesives, including **nanoaggregates**, **nanocrystals**, and **nanofibrils**, driven by the hierarchical hydrogen-bonding interactions. At the molecular level, hydrogen bonding drives the formation of nanoaggregates. At the nanoscale, the adipic acid blocks crystallize into nanocrystals, while the aligned oligomers form nanofibrils. This multi-level organization enables **efficient energy dissipation**, **crack resistance**, and **thermal tolerance**, ensuring that the adhesive maintains its mechanical properties and adhesion strength even under extreme conditions.

(3) **Dual Control of Properties.** Our strategy simultaneously optimizes both the **mechanical properties** (e.g., strength, toughness) and **thermal stability** of the adhesives, which is a significant departure from conventional approaches that often focus on one aspect at the expense of the other.

(4) **Advancements Beyond Literature Reports.** While we build on established principles of hydrogen bonding, our work extends these concepts by demonstrating how processing conditions (e.g., **solution-shearing**) can be leveraged to tailor material properties in a way that has not been previously reported. This approach offers a **new framework** for designing adhesives with enhanced thermal and mechanical performance. Our work demonstrates a significant enhancement in the performance of hydrogen-bonded thermal-resistant adhesives, as highlighted in Table S7 of the revised SI.

(5) **Scalability and Versatility.** The solution-shearing process is highly scalable and compatible with a wide range of substrates. Oligomers, with their lower viscosity and better solubility, enhance their processability, making this approach practical for various applications.

Mechanistic Clarity and Data Depth

We have made a detailed discussion and additional characterization in mechanistic explanation:

(1) A more in-depth discussion of the hydrogen bonding in hierarchical nanostructures and how they contribute to the mechanical strength, toughness, and thermal-resistant adhesion.

As stated in paragraph 2, page 6 of the revised manuscript,

“In this work, the solution-sheared supramolecular oligomers featured the hierarchical nanostructures formed through hydrogen bonding. These nanostructures include nanoaggregates, nanocrystals, and nanofibrils, which play distinct yet complementary roles in enhancing bulk cohesion and interfacial adhesion: (1) Nanoaggregates are self-assembled clusters of oligomers driven by strong hydrogen-bonding interactions, particularly the quadruple hydrogen bonds formed by UPy units. The nanoaggregates create a network of physical crosslinks within the adhesive, improving energy dissipation and crack resistance. (2) Nanocrystals are ordered, crystalline domains formed by the adipic acid units within the oligomers. The solution-shearing process promotes the growth and alignment of nanocrystals, enhancing the adhesive's mechanical strength and thermal stability. (3) Nanofibrils are elongated, fiber-like structures that are oriented parallel to the direction of shear. These nanofibrils enhance the adhesive's mechanical robustness and toughness by dissipating stress during deformation and providing structural integrity. The synergistic effect of nanoaggregates, nanocrystals, and nanofibrils, stabilized by hierarchical hydrogen bonding, enables the adhesives to achieve unprecedented performance in terms of strength, toughness, and thermal resistance.”

(2) The characterization techniques in the bulk phase, including DSC, XRD, AFM, SEM, and SAXS, to better illustrate the **oriented hierarchical nanostructures** and their role in bulk cohesion.

- DSC Analysis: an increase in melting temperatures and crystallinity at higher shear rates, indicating the formation of nanostructures (Figure 3a of the revised manuscript).
- XRD and SAXS Results: an enhancement in crystallinity from 43% to 60% and significant growth in the sizes of **nanocrystals**, validating our claims about the impact of shearing on nanostructural properties (Figure 3b, 3e-g of the revised manuscript).

- AFM phase images show the **aligned nanofibrils** of SST-2-10 (Figure 3c-d of the revised manuscript).
 - SEM images indicate the formation of **microfibers** of SST-2-10 (Figure S14 of the SI).
- (3) The surface characterization, including Temperature-dependent ATR-FTIR and Raman, AFM-based indentation:
- ✧ Temperature-dependent ATR-FTIR analysis confirmed the presence of C=O and NH units at the surface of SST-2-10 (Figure S18 of the SI).
 - ✧ Raman spectroscopy and mapping revealed **higher intensities of dissociated UPy motifs and ordered urea units** in sheared samples compared to unsheared ones, suggesting stronger interfacial bonding (Figures 4f-g of the manuscript).
 - ✧ Peak Force Quantitative Nanomechanical Mapping indicated a **higher surface modulus and improved adhesion force** of SST-2-10 in contrast to the control sample, confirming the role of hydrogen-bonded nanocrystals (Figure 4h-i).

Figure R30. As a copy of Figure 1 showed that chemical structures and the synthetic route of supramolecular thermoplastic oligomers (a); Interfacial adhesion and bulk cohesion enabled by the solution-shearing of supramolecular thermoplastics (SST-n-m) that display ordered and enlarged nanocrystals and aligned nanofibrils in the bulk phase and anchored to the surface (b); Multiscale engineering used for the design and construction of supramolecular structural adhesives with high-strength and thermal-resistant adhesion (c).

Figure R31. As a copy of Figure 3 of the revised manuscript showed structural properties. (a) DSC curves of SST-2-m treated with the shear rate of 0, 5, 10, and 15 mm/s. (b) XRD profiles of SST-2-m on a glass plate and treated with the shear rate of 0, 5, 10, and 15 mm/s. The curves were recorded by the second heating scan from -60 to 180°C with a heating rate of 10°C/minutes. AFM phase images of the nanofibrils arrangement of SST-2-0 (c) and SST-2-10 (d) on the glass substrate. (e) The scattering intensity versus q vector parallel to nanofibrils of the shearing direction. (f) The scattering intensity versus q vector perpendicular to nanofibrils of the shearing direction. (g) The average distance between adjacent nanocrystals of sheared films parallel and perpendicular to the shearing direction at a shear rate of 0, 5, 10, and 15 mm/s. (h) The cross-sectioned schematic of the average distance between adjacent nanocrystals before and after shearing process.

Figure R32. As a copy of Figure S14 shows the formation of microfibrils of SST observed in SEM image.

Figure R33. As a copy of Figure 4 shows **Bulk and interfacial characterizations**. (a) Tensile stress-strain curves of SST-n-m recorded with a stretching rate of 10 mm/minutes. The tensile performance shown in this figure were performed in a direction parallel to the nanofibrils. Bar charts of the Young's modulus (b) and toughness (c) of SST-n-m. (d) Stress relaxation tests of SST-2-10 measured at different temperature. (e) The fitted curve of natural logarithm relaxation time versus temperature for SST-2-10. (f) Intensities of the disassociated UPy motifs and urea units tested by raman spectra and (g) mapping on the surface layers of SST-2-0 and SST-2-10. The surface adhesion force distributions and mapping of SST-2-0 (h) and SST-2-10 (i). (j) The improved and fracture-resistant cohesion is attributed to high energy ordered nanostructures that effectively pin cracks within the bulk. (k) High interfacial adhesion arises from anchored nanocrystals, which pin cracks both at the interface and within the bulk. These ordered nanostructures require higher energy for crack propagation compared to amorphous chains, thereby enabling fracture-resistant cohesion and adhesion. All the error bar represent the SD with at least three replicates.

Figure R34. As a copy of Figure S18 shows that stretching bands of C=O and N-H at the adhesive surface with the temperature changes.

Figure R35. As a copy of Figure S19 shows surface modulus measurements of SST-2-0 and SST-2-10, tested by AFM-based indentation.

(5) Experiments (e.g., stress relaxation, rheology, and viscosity) to quantify the **reversibility** and **dynamics** of the hydrogen-bonding interactions under varying temperature conditions.

Figure R36. As a copy of Figure 4d-e shows stress relaxation tests of SST-2-10 measured at different temperature.

Figure R37. As a copy of Figure 5d-e shows the temperature sweep rheological analysis of SST-2-10 at a constant frequency of 1.0 rad/s from 30 to 180 °C and variation of SST-2-10 with temperature between 120 to 180 °C measured during heating and cooling.

Impact and Urgency:

The key discovery of this work lies in the development of a **solution-shearing** strategy that enables the **growth and orientation of nanocrystals** and the **alignment of nanofibrils** in hydrogen-bonded supramolecular oligomers. Importantly, this strategy simultaneously controls both the **adhesion chemistry at the interface** and the **nano/microstructure of the adhesive layer**. This dual control enables a **synergistic enhancement** in both **bulk cohesion** and **interfacial adhesion**, which represents a significant departure from conventional approaches that often optimize these properties separately.

Furthermore, our work establishes a new framework for designing **thermal-resistant hydrogen-bonded adhesives** with tailored properties, offering insights into the interplay between **processing conditions, material structure, and performance**. These findings are not only relevant to adhesive applications but also provide broader implications for the design of advanced functional materials.

We believe this work not only has strong **application-focused elements**, benchmarking against commercial and previously reported adhesives, but also presents the **fundamental scientific principles** and **engineering strategies**, which are equally impactful and broadly relevant to the materials science community.

Also, thank you for your careful observation and for raising this important point regarding the removal of residual DMSO during the solution-shearing process. We appreciate the opportunity to clarify this aspect of our experimental procedure. While the boiling point of DMSO is indeed 189°C, it is important to note that we made specific treatments in our experimental procedure:

(1) The initial heating at 160°C for 30 minutes ensures the majority of the DMSO is evaporated, as the temperature is close to its boiling point and sufficient to drive off most of the solvent.

(2) The subsequent heating at 60°C for 12 hours is intended to remove any trace amounts of residual DMSO. Although 60°C is below the boiling point, prolonged heating at this temperature can effectively remove residual solvent due to the continuous application of heat and the large surface area of the film.

To confirm the complete removal of DMSO, we performed FTIR and TGA analysis on the SST films:

(1) FTIR Analysis. The absence of characteristic peaks in the FTIR spectrum of the SST film, especially the peaks for DMSO at 1024 cm⁻¹ assigned to stretching vibration, and at 950 cm⁻¹ for bending vibration, confirms that no detectable DMSO remains.

Figure R38. As a copy of Figure S22 of the SI shows that FTIR spectrum of the sheared sample before and after heating at 60°C for 12 hours.

(2) TGA Analysis. TGA curves of the oligomers showed no significant weight loss below 200°C. Besides, the TGA curves of SST-2-0 and SST-2-10 nearly overlap. These experimental results confirm the absence of residual DMSO solvent.

Figure R39. As a copy of Figure S23 of the SI shows that TGA curves of SST-2-0 and SST-2-10.

To avoid any ambiguity, we have revised the experimental section to more clearly describe the solvent removal process and included the supporting data from FTIR and TGA, as shown below:

“The SST solution was prepared in DMSO at a concentration of 10 mg/mL and then heated in a glass dish at 160°C for 30 minutes to evaporate the majority of the solvent. The resulting sheared film was subsequently left on the heating plate at 60°C for 12 hours to ensure the complete removal of any residual DMSO. The absence of residual DMSO was confirmed by FTIR and TGA analysis (Figure S22-23).”

We believe that the **novel processing strategy** and **exceptional performance** of our adhesive, which significantly **surpasses existing benchmarks**, present a compelling case for its publication in a high-impact journal. Our work not only introduces a groundbreaking approach to adhesive design but also demonstrates unprecedented performance metrics that address critical challenges in the field. These advancements have the potential to inspire new research directions and applications across multiple disciplines, including materials science, biomedical engineering, and industrial manufacturing. We are confident that the broad implications and transformative nature of our findings will resonate strongly with the diverse readership of *Nature Communications*, fostering interdisciplinary interest and driving further innovation

Thank you again for your valuable feedback, which helps us improve the clarity, depth, and impact of our manuscript. We hope the revisions have addressed your concerns and demonstrated the urgency and significance of our work for the broader scientific community.

Response to the reviewers' comments

Reviewer #4:

In this review comment, I mainly focus on the author's rebuttal and revision in response to flagged technical concerns. The authors have addressed majority comments from Reviewer 1 and Reviewer 2. However, for the concerns on inadequate novelty and unclear mechanism raised by Reviewer 3, I don't think the authors have effectively addressed these two concerns yet.

Response:

We sincerely appreciate the reviewer's careful evaluation and valuable feedback. We have made a concerted effort to more explicitly articulate the originality of our approach and to clarify the underlying mechanisms with additional evidence and analysis. Below, we provide detailed responses to each concern raised by the reviewer.

1. In the response letter, the authors claim that “this dual control enables a synergistic enhancement in bulk cohesion and interfacial adhesion, which has not been achieved in prior studies”, which is true. However, the authors didn't provide data to show how their approach synergistically enhances both bulk cohesion and interfacial adhesion. To make this mechanism clear, the authors are recommended to measure interfacial fracture toughness and interfacial fatigue threshold, which can help to decouple these two mechanisms.

Response:

We thank the reviewer for highlighting the importance of mechanistically decoupling bulk cohesion and interfacial adhesion. We fully agree that quantitative metrics such as interfacial fracture toughness and fatigue threshold are essential to substantiate the proposed synergistic effect. To address this, we have conducted standardized 90° peeling tests under both single-cycle and cyclic loading conditions, following established protocols (Nat. Commun. 2020, 11, 10171; Sci. Adv. 2019, 5: eaau8528; PNAS 2019, 116, 10244-10249)

Interfacial Fracture Toughness: This metric reflects the energy required to propagate a crack along the adhesive-substrate interface. We have measured the interfacial fracture toughness of SST-2-10, which yielded a value of 6700 J/m², significantly higher than the unsheared sample SST-2-0 (1800 J/m²), as shown in Figure R1a. The 3.7-fold increase confirms the enhanced interfacial adhesion due to hydrogen-bonded nanocrystals anchored at the interface.

Interfacial fatigue Threshold: This defines the critical energy release rate below which cracks cease to grow under cyclic stress. As shown in Figure R1b, cyclic peeling tests revealed a fatigue threshold of 1360 J/m² for SST-2-10, compared to 230 J/m² for SST-2-0. The 5.9-fold improvement demonstrates that the aligned nanofibrils within the bulk matrix effectively dissipate energy and resist fatigue crack growth.

Figure R1. The measured interfacial fracture toughness (a) and interfacial fatigue threshold (b) for SST-2-10 and SST-2-0 on stainless steel substrates.

The decoupled metrics now clearly show how our approach enables dual control: interfacial adhesion is enhanced by surface-anchored nanocrystals, while bulk cohesion is reinforced by aligned nanofibrils.

These results are now included in Figure S20 and discussed in the Results section at page 12 of the revised manuscript,

“To further clarify how the solution-shearing process synergistically enhances both bulk cohesion and interfacial adhesion, we quantified interfacial fracture toughness and fatigue threshold via 90° peel testing on stainless steel substrates under a single cycle and multiple cycles of loads. As indicated in Figure S20, SST-2-10 yielded an interfacial fracture toughness value of 6700 J/m², 3.7-fold higher than SST-2-0 (1800 J/m²), which confirms that solution-shearing enhances interfacial adhesion through hydrogen-bonded nanocrystals anchoring. Further, cyclic peeling tests revealed an interfacial fatigue threshold of 1360 J/m² for SST-2-10, compared to 230 J/m² for SST-2-0, demonstrating that aligned nanofibrils in the bulk phase resist crack propagation and effectively dissipate energy during repeated loading. Together, these decoupled metrics validate the dual-control mechanism underpinning our synergistic design.”

We hope this additional analysis satisfactorily addresses the reviewer’s concern.

2. The authors show that the toughness and lap-shear strength of the proposed SST outperform existing thermal-resistant adhesives including commercially available epoxy-based adhesives. The summarized strength of epoxy-based adhesives in the manuscript is typically below 20 MPa as summarized in Figure 2 and Figure 5, which can be confusing. In fact, the existing commercially available epoxy-based adhesives work pretty well, which can achieve strength up to 70 MPa (Rudawska, Anna. “Mechanical properties of selected epoxy adhesive and adhesive joints of steel sheets.” *Applied Mechanics* 2, no. 1 (2021): 108-126.). Thus, it is not convinced that the proposed SST truly outperforms epoxy-based adhesives.

Response:

We appreciate the reviewer for the thoughtful comment and for drawing attention to the performance benchmark of epoxy adhesives. We respectfully address the concern in the following three points:

(1) Clarification on Epoxy Adhesive Strength

The mentioned study by Rudawska (2021) reports a compressive strength of 70 MPa for epoxy adhesives, rather than lap shear strength. Their actual lap shear strength ranges from 6.7-13.3 MPa, which is consistent with our experimental data of commercial epoxy adhesive (e.g., Loctite Quick Set), tested under identical testing conditions as our SST-2-10 samples (Figures 2h and 5g).

(2) Beyond Lap Shear Strength—Multifunctional Advantages of SST Adhesives

While we acknowledge that some high-performance epoxy adhesives can reach lap shear strengths of 10-40 MPa when optimized and applied with surface treatment (e.g. *Angew. Chem.* 2024, 63, e202408840; *Adv. Mater.* 2023, 2300802; *ACS Mater. Lett.* 2021, 3, 1003-1009), our SST-2-10 achieves a lap shear strength of 30.6 MPa on untreated stainless-steel substrates, demonstrating competitive strength with far simpler processing. More importantly, SST adhesives exhibit distinct functional advantages not attainable with conventional epoxies:

- **Thermal Stability:** SST-2-10 maintains a lap shear strength of 21 MPa at 120°C (Figure 5g), whereas Loctite Quick Set drops to 2 MPa at 95°C (Table S7). Most epoxy adhesives degrade significantly above 80°C.
- **Reversibility:** Unlike permanently crosslinked epoxies, SST-2-10 can undergo over 20 rebonding cycles (Figure 2f), whereas epoxies are irreversible.
- **Reusability:** The adhesion strength remains consistent after solution-shearing reprocessing (Figure 2f and S17)

(3) Positioning SST Adhesives in the Broader Context:

This work aims not merely to exceed absolute strength benchmarks, but to introduce a new class of

thermally resistant, reprocessable adhesives that combine robust performance (30.6 MPa lap shear) —while adding unique functionalities. We have revised the text to clarify this balance, as shown in Discussion section at page 16,

“While optimized epoxy adhesives can exhibit higher lap shear strength (10-40 MPa) under ideal conditions and surface treatment, the SST platform represents a significant shift: offering competitive adhesion on untreated surfaces, combined with thermal stability, reusability, and reversible adhesion—functionalities unattainable in conventional epoxy systems. These features render SST adhesives particularly suitable for emerging applications requiring repeated bonding, thermal resilience, and sustainable material use.”

We hope this response addresses the reviewer’s concerns and clarifies the unique positioning of SST adhesives relative to conventional systems.

Figure R2. As a copy of Figure 2f and 2h shows that lap shear strength of SST-2-10 on aluminium substrate over 20 cycles and benchmarking with commercial adhesives on diverse substrates.

Figure R3. As a copy of Figure 5g shows the comparisons of lap shear strength between existing thermal-resistant adhesives and this study at elevated temperature.

Figure R4. As a copy of Figure S17 shows tensile stress curves of the reprocessed SST-2-10.

3. The authors attribute the thermal-resistant adhesion to the stability of ordered nanostructure at elevated temperature. Since the melting temperatures of most crystalline polymers are typically above 150 °C, it is hard to understand how stable adhesion performance across a temperature from 20 to 120 °C by SST outperforms existing polymer-based adhesives.

Response:

We appreciate the reviewer's thoughtful question regarding the thermal stability mechanism of SST adhesives. We agree that most crystalline polymers exhibit melting points above 150 °C and tend to lose mechanical integrity sharply near those temperatures. However, the SST system differs fundamentally in both structure and transition behavior. Below, we elaborate on the key mechanisms supporting SST's high adhesion performance across a wide temperature range (20–120 °C):

(1) Gradual Thermal Transition

While the melting point of SST-2-10 is ~155°C (Figure 3a), its thermal resistance operates through a gradual stability gradient rather than a binary crystalline–amorphous transition. At 90°C, the crystallinity remained above 50%, resulting in a lap shear adhesion strength exceeding 25 MPa. At 120°C, high viscosity (over 10⁴ Pa·s) preserves 21 MPa. Even at 130°C, with residual crystallinity above 20%, the adhesion strength remains above 15 MPa (Figure 5b and S21). This progressive reduction in crystallinity is distinct from conventional polymer adhesives that typically fail near their melting point due to abrupt loss of molecular ordering.

(2) Dynamic Bond Network for Stress Relaxation

The UPy-based supramolecular networks provide temperature-adaptive behaviour. Below 100°C, stable H-bonds maintain structural integrity. Above 100°C, controlled bond dissociation allows stress relaxation without interfacial or cohesive failure (Figure 4d).

(3) Multiscale synergy

Our solution-shearing technique induces a multiscale structural hierarchy that collectively enhances thermal adhesion performance:

- **Surface-anchored nanocrystals** provide robust interfacial binding;
- **Bulk-aligned nanofibrils** enhance mechanical integrity and resist crack propagation;
- **Dynamic hydrogen bonds** allow for molecular rearrangement and energy dissipation under thermal stress.

Together, these mechanisms enable SST adhesives to maintain high performance at temperatures where conventional adhesives either soften irreversibly or undergo catastrophic failure.

To improve clarity, we have added further clarification to emphasize this graduated stability mechanism and its advantages over existing polymer paradigms, as shown in “Thermal-Resistant Adhesion” at page 14 of the revised manuscript,

“Figure 5a reveals that SST oligomers maintain stable adhesion at elevated temperature through temperature-adaptive nanostructures: high crystallinity (>50% at 90°C) provides mechanical integrity, while dynamic hydrogen bonding networks enable stress relaxation above 100 °C. Both features demonstrate the ability of SST oligomers to maintain mechanical performance through gradual thermal transitions, in contrast to conventional adhesives that suffer abrupt property degradation near their melting points.”

“The solution-shearing-engineered hierarchy—combining (i) interfacial nanocrystal anchors, (ii) bulk-aligned nanofibrils, and (iii) dynamic bond reconfiguration—explains the exceptional retention of 21 MPa adhesion at 120 °C, where most polymer adhesives experience dramatic performance degradation.”

We hope this explanation and the newly included experimental data adequately address the reviewer's concern.

Figure R5. As a copy of Figure 3a shows the melting point of SST adhesives.

Figure R6. As a copy of Figure 4d shows the stress relaxation curves of SST-2-10 at different temperature.

Figure R7. As a copy of Figure 5b shows the lap shear strength of SST-2-10 tested at different temperature.

Figure R8. As a copy of Figure S21 shows the crystallinity of SST-2-10 at different temperature.

4. The authors provided detailed information on thermal-resistant adhesion measurement following ASTM D1002, which is great. However, the explanation of the “thermal-resistant adhesion” test is unclear. The text states that samples were equilibrated in a heated chamber, but the actual shear tests were performed at room temperature. If the goal is to evaluate high- temperature adhesion, shear testing should be conducted under elevated temperatures, or the rationale for pre-heating followed by room-temperature testing must be clarified.

Response:

We appreciate the reviewer's careful attention to our experimental methodology. We would like to clarify our high-temperature adhesion testing protocol:

(1) Testing Conditions

All lap shear adhesion tests at elevated temperatures were conducted isothermally at the specified temperatures (ranging from 20 to 160 °C) using an environmental chamber equipped on the INSTRON-5566 testing system. Specifically:

- Samples were equilibrated at the target temperature for 10 minutes to ensure uniform thermal distribution across the adhesive joint
- Mechanical testing was performed in situ at the target temperature without any cooling phase, in full accordance with the ASTM D1002 standard

(2) Protocol Clarifications

We have revised the Methods section (Page 19) to explicitly state:

"High-temperature lap shear tests were performed isothermally at each target temperature (20-160°C). Bonded samples were pre-equilibrated in an environmental chamber for 10 minutes to reach thermal equilibrium before mechanical testing. Shear testing was carried out directly at the test temperature using an INSTRON-5566 equipped with a temperature-controlled chamber, with a constant loading rate of 10 mm/min in accordance with ASTM D1002."

5. The strategy using solution-sheared supramolecular oligomers can be a novel angle due to its advantage in structural control, processability, adhesive performance, and scalability. However, the authors didn't specifically compare this strategy to traditional polymer-based adhesive fabrications.

Response:

We are grateful to the reviewer for raising this important point. Indeed, the use of solution-sheared supramolecular oligomers represents a departure from conventional polymer-based adhesive fabrication methods. Below, we provide a detailed comparison highlighting why this strategy offers superior control over structure formation, processability, and ultimately, adhesion performance—particularly through a direct contrast with representative polymer systems.

1) Precise Structural Control: The manuscript highlights the ordered nanostructures formed via solution shearing of oligomers, but does not clarify whether similar effects are achievable in polymers, or why polymers are less favorable in this context.

Response:

We appreciate the reviewer's comment and the opportunity to elaborate on why solution-sheared supramolecular oligomers offer superior structural control compared to traditional polymers. This advantage arises from two fundamental differences: **minimal chain entanglement** and **dynamic bonding**, both of which facilitate ordered nanostructure formation during the shearing process. Below, we provide direct experimental comparisons between our SST oligomers and conventional polymers (Polyurethane (PU) Adhesive (Mn = 80 kDa) and Polypropylene (PP) thermoplastic material (Mn = 30 kDa)).

Experimental Comparison (Figure R9-10):

Parameter	SST-2-10 (2.4 kDa)	PU adhesive (80 kDa)	PP polymer (30 kDa)
-----------	--------------------	----------------------	---------------------

Nanocrystal size (SAXS)	10.5 nm	4.2 nm	5.8 nm
Adhesion force (AFM)	9.1 nN	2.3 nN	3.1 nN

Key Differences:

- **Minimal Chain Entanglement Enables Uniform Alignment:**

Rheological measurements (Figure R11) confirm SST-2 oligomer ($M_n = 2.4$ kDa) enables minimal chain entanglement (No G' plateau modulus); while for conventional polymers (PU/PP ($M_n > 30$ kDa)), extensive chain entanglements are evidenced by G' plateau modulus of PU (3.8×10^5 Pa) and PP (1.7×10^5 Pa), which disrupts chain mobility and inhibits uniform alignment during shearing, thus compromising nanocrystal ordering.

- **Dynamic Bonding Promotes Reorganization:**

SST's dynamic H-bond motifs enable reversible reorganization during shearing, as confirmed by variable-temperature FTIR (Figure S18); whereas PU's crosslinks and PP's chain rigidity prevent reorganization and permanently lock chain conformations.

These findings highlight that oligomeric systems are uniquely capable of forming highly ordered nanostructures via solution shearing, owing to their low entanglement and dynamic supramolecular interactions. In contrast, high-molecular-weight polymers inherently limit such structural precision due to entanglement and rigidity. This mechanistic distinction underscores the rationale for adopting an oligomer-based strategy in our adhesive design.

Figure R9. The gyration radius of nanocrystals in SST-2-10 and conventional polymers. All the tests used identical substrates, solvent, and concentration.

Figure R10. Adhesion force distribution of SST-2-10 and conventional polymers.

Figure R11. Frequency sweep of SST-2-10 and conventional polymers tested at 180°C.

Figure R12. As a copy of Figure S18 shows temperature-dependent ATR-FTIR spectra for C=O and N-H stretching vibration.

2) Improved Processability: The authors attribute better processability to the intrinsic features of oligomers (lower viscosity, reduced entanglement), but no data or comparison with polymers is provided. It also remains unclear whether these features are general or specific to their design.

Response:

We thank the reviewer for raising this important point. To substantiate our claim regarding enhanced processability, we conducted direct rheological and processing comparisons between our SST oligomer and two representative high-molecular-weight polymers. All tests were performed under identical conditions, including substrates, solvent, and solution concentration.

(1) Rheological comparison (180°C)

Property	SST-2-10 (2.4 kDa)	PU adhesive (80 kDa)	PP polymer (30 kDa)
Zero-shear viscosity	10 Pa·s	6200 Pa·s	930 Pa·s
G' plateau modulus	None	3.8×10^5 Pa	1.7×10^5 Pa

Interpretation: SST-2-10 displays a 90-600× lower viscosity than PU and PP at the same temperature (Figure R11), with no measurable G' plateau, indicating negligible entanglement. In contrast, the plateau moduli of PU and PP confirm extensive chain entanglement, which hampers shear-induced alignment and film uniformity.

(2) Processability outcomes

Parameter	SST-2-10 (2.4 kDa)	PU adhesive (80 kDa)	PP polymer (30 kDa)
Minimum shear rate for alignment	10 mm/s	>500 mm/s	>100 mm/s
Thickness variation	±5%	±35%	±25%

Key Findings:

Due to low viscosity and entanglement-free chains, SST-2-10 enables uniform film formation at shear rates as low as 10 mm/s, compared to >500 mm/s required for PU. Furthermore, SST coatings exhibit significantly improved thickness uniformity (±5%) versus PU and PP (±35% and ±25%, respectively), indicating more consistent deposition during processing.

These results demonstrate that the processability advantages of our SST system stem from intrinsic features of the oligomer design—specifically, low molecular weight, absence of chain entanglement, and dynamic bonding. This facilitates low-energy, high-uniformity processing, making SST especially suitable for scalable fabrication methods such as roll-to-roll manufacturing without the need for high-shear or thermal activation commonly required by conventional polymers.

Figure R13. Viscosity of conventional polymers at 180°C.

3) Scalability: The authors state that the process is scalable, yet present no experimental evidence or process discussion related to large-area coating, process uniformity, or industrial compatibility. All data appears limited to lab-scale, static shear coating.

Response:

We thank the reviewer for highlighting the importance of scalability in translating laboratory innovation to industrial application. While this study is focused on establishing fundamental structure–property relationships, we agree that scalability is a critical dimension for practical adoption. Below, we clarify our current positioning and the rationale behind our approach.

(1) Fundamental Focus of This Work

This work is intended as a proof-of-concept that elucidates the molecular and nanoscale mechanisms underlying the performance of supramolecular oligomer adhesives—including dynamic bonding, nanocrystal formation, and shear-induced alignment. Lab-scale solution-shearing was employed to achieve precise control over shear rates, film thickness, and structural characterization at the microscale and nanoscale. These insights are essential prerequisites to inform successful scale-up.

(2) Oligomer’s Scalability Potential over Conventional Polymers

- Low-Viscosity Processing: Oligomers (e.g., SST-2-10, $M_n = 2.4$ kDa) exhibit $\sim 100\times$ lower viscosity than high-MW polymers (PU/PP) at processing temperatures, allowing compatibility with roll-to-roll (R2R) or slot-die coating.
- Rapid Fabrication: Solution-shearing aligns nanofibrils in seconds (vs. hours for polymer crystallization), critical for high-throughput manufacturing.

(3) Industrial Compatibility Pathways

We acknowledge the reviewer's request for empirical scalability demonstrations. As part of our ongoing research, we are initiating the following next steps:

- Pilot-Scale Testing: In collaboration with industrial partners, we are adapting the SST formulation for continuous coating over 30–50 cm substrate widths at linear speeds of 1–5 m/min.
- Process Optimization: Parameters (shear rate, temperature) will be adapted for continuous flow, leveraging the oligomer's low entanglement.

We have added the following text to the Discussion Section (page 17) to contextualize scalability:

"Though the present study focuses on fundamental material mechanisms using lab-scale shear coating, the SST oligomer system is intrinsically compatible with scalable processing. The combination of low viscosity, minimal entanglement, rapid nanostructure formation, and melt-processability positions it well for integration into roll-to-roll or slot-die coating workflows. Future work will quantify large-area performance, leveraging these insights to bridge the gap between laboratory innovation and industrial deployment."

We hope this response clarifies our position and commitment to validating the scalability of the proposed adhesive system.